# Test-Time Debiasing with Probabilistic Prompts
# via Wasserstein Distance in Vision-Language Models

Chengye Wang [1]    Yuyuan Li [2]    Xiaohua Feng [1]    Xiaolin Zheng [1,*]    Chaochao Chen [1]

## Abstract

Vision-Language Models (VLMs) inherit social biases from large-scale pretraining data, and these biases can amplify in downstream tasks, leading to systematic performance disparities across sensitive groups. Due to the high training cost and the risk of catastrophic forgetting, recent research has focused more on lightweight *test-time* debiasing, aiming to obtain an ideal fair embedding for each query. However, such point-based corrections are often unstable and become notably weaker in multi-class settings, where group structure cannot be adequately captured by a single point. Therefore, we propose W4D, a distributional debiasing framework that reframes fairness as aligning query embedding distributions to group reference distributions under the Wasserstein distance, which provides a geometry-aware notion of discrepancy beyond mean shifts. To make this alignment practical at test-time, W4D introduces probabilistic prompts that induce controlled distributional perturbations and optimizes a Wasserstein-based objective to reduce cross-group disparity while preserving task-relevant semantics. This distributional perspective improves robustness in multi-class debiasing and yields a stronger fairness–utility trade-off across diverse VLM downstream evaluations. Our code is available at https://github.com/QDRhhhh/W4D.

## 1. Introduction

Vision-Language Models (VLMs) have introduced a new paradigm for multimodal understanding and have achieved impressive performance on a wide range of downstream tasks such as zero-shot classification (Radford et al., 2021; Alayrac et al., 2022) and text-to-image (Yao et al., 2021; Dall'Asen et al., 2024) retrieval. By jointly modeling visual and textual inputs in a shared embedding space, these models are able to generalize to novel concepts and compositions without task-specific supervision (Girdhar et al., 2023; Herzig et al., 2023; Zhou et al., 2024). However, since VLMs are pretrained on large-scale datasets that inevitably contain socially biased samples, they often inherit or even amplify biases related to sensitive social attributes, including gender, race, and age (Silva et al., 2021; Wang et al., 2022; Janghorbani & De Melo, 2023). Such biases can manifest as stereotypical associations, disparate performance across demographic groups, or systematically skewed similarity scores. These issues can propagate into downstream tasks such as zero-shot classification, leading to more severe racial and gender discrimination in model behavior and performance (Hall et al., 2023; Dehdashtian et al., 2024).

To mitigate such bias in VLMs, early research primarily proposes *training-based methods*. These methods aim to identify biased patterns within the model and reduce their impact by fine-tuning the model's parameters and adjusting its representations (Zhang & Sang, 2020; Zhu et al., 2023; Si et al., 2023; Luo et al., 2024). For example, to mitigate gender bias in engineer representation, they train the model to maximize its reliance on occupational labels while minimizing its dependence on gender information, thereby achieving debiasing (Wang et al., 2021). However, due to the large parameter scale of VLMs, updating these models often leads to a significant increase in training costs. Furthermore, extensive parameter adjustments can not only trigger catastrophic forgetting but also lead to performance degradation on other tasks, presenting substantial challenges in practical applications (Mukhoti et al., 2024).

To solve these issues, recent research has increasingly focused on **test-time methods** that adjust the model's output embeddings without updating model parameters (Chuang et al., 2023; Zhao et al., 2025; Zhang et al., 2025). Some straightforward methods directly remove certain attribute-specific dimensions from the image embeddings (Jung et al., 2024; Zeng et al., 2025). However, these methods lack formal definitions and theoretical grounding, limiting their

[1] College of Computer Science and Technology, Zhejiang University, Hangzhou, China [2] School of Communication Engineering, Hangzhou Dianzi University, Hangzhou, China. Correspondence to: Xiaolin Zheng <xlzheng@zju.edu.cn>.

*Proceedings of the 43rd International Conference on Machine Learning*, Seoul, South Korea. PMLR 306, 2026. Copyright 2026 by the author(s).

interpretability and making it hard to ensure complete removal of the target attribute. Consequently, further methods define VLMs fairness by enforcing equal average distances between the text embedding and image samples across categories (Gerych et al., 2024; Han et al., 2026; Li et al., 2024). Under this constraint, an ideal debiased embedding point is first derived, yielding a closed-form solution for binary attributes. Through constrained optimization, the original text embedding is encouraged to align with the ideal debiased embedding point. However, as shown in the upper-left panel of Figure 1, these methods still suffer from the following two limitations:

- **Insufficiency of point-based debiasing.** Previous methods define a single debiased embedding point, which fails to capture these distributional differences and ensure fairness across samples.

- **Limited support for multi-class attributes.** Most existing methods are designed for binary attributes, with limited consideration and analysis of multi-class attributes, causing weaker debiasing performance.

Therefore, we propose W4D (Wasserstein-for-Debias), a novel test-time debiasing method based on Wasserstein distance for multi-class sensitive attributes that operates without modifying any model parameters. Our key insight is that fairness issues in VLMs are inherently *distributional*: different attribute groups often form distinct, structured clusters in the embedding space, so debiasing by solving for a single ideal point can only correct the mean behavior while leaving substantial sample-level disparities untouched. Accordingly, W4D reframes debiasing as aligning the model output to *group distributions* and uses a geometry-aware distance to compare distributions, which provides a principled way to account for intra-group spread and inter-group structure rather than collapsing them into a point estimate. This perspective also yields a natural extension to multi-class attributes: instead of relying on binary-specific closed forms, we directly encourage *uniform closeness* across all groups by minimizing the disparity of group-wise distances, ensuring that no attribute class is systematically advantaged or disadvantaged. As a result, W4D offers a lightweight test-time solution that scales to multi-class sensitive attributes and empirically improves debiasing effectiveness while maintaining comparable downstream performance.

In summary, our main contributions are:

- We introduce a new *distributional* perspective on debiasing in VLMs with multi-class sensitive attributes, providing formal mathematical definitions and theoretical analysis.

- We propose W4D, a test-time debiasing method based on Wasserstein distance and probabilistic prompts, enabling lightweight, stable, and effective debiasing in the multi-class setting.

- We conduct extensive experiments across multiple datasets and baselines, demonstrating the effectiveness, efficiency, and practicality of our proposed method.

## 2. Preliminary

### 2.1. Wasserstein Distance

Wasserstein distance (Givens & Shortt, 1984) is used to measure the discrepancy between distributions corresponding to different attributes Let $(\mathcal{X}, d)$ be a metric space and let $\mu, \nu \in \mathcal{P}(\mathcal{X})$ have finite $p$-th moments for some $p \geq 1$. The ground cost is defined as $c(x, y) = d(x, y)^p$.

**Monge formulation.** Monge (Monge, 1781) optimal transport searches for a deterministic map $T : \mathcal{X} \to \mathcal{X}$ that pushes $\mu$ to $\nu$ (denoted $T_{\#}\mu = \nu$):

$$W_p^p(\mu, \nu) = \inf_{T : T_{\#}\mu = \nu} \int_{\mathcal{X}} d(x, T(x))^p \, d\mu(x).$$

This problem is generally *non-convex* due to the hard constraint $T_{\#}\mu = \nu$ and the restriction to deterministic maps, and a minimizer may fail to exist in practical settings.

**Kantorovich formulation.** To obtain a convex relaxation that allows mass splitting, Kantorovich optimizes over couplings (Kantorovitch, 1958). Let $\Pi(\mu, \nu)$ denote the set of joint measures $\pi$ on $\mathcal{X} \times \mathcal{X}$ with marginals $\mu$ and $\nu$, i.e., $\pi \in \Pi(\mu, \nu)$ iff $P_1\pi = \mu$ and $P_2\pi = \nu$, where $P_1, P_2$ are the marginalization operators. Then

$$W_p^p(\mu, \nu) = \inf_{\pi \in \Pi(\mu, \nu)} \int_{\mathcal{X} \times \mathcal{X}} d(x, y)^p \, d\pi(x, y),$$

and the distance is defined as

$$W_p(\mu, \nu) = \left( W_p^p(\mu, \nu) \right)^{1/p}.$$

### 2.2. Notation.

Let $\{(I_i, a_i, c_i)\}_{i=1}^n$ denote labeled samples drawn from a joint distribution over images and attributes, where $I_i$ is an image, $a_i \in \mathcal{A}$ is a *sensitive* attribute (e.g., race or gender), and $c_i \in \mathcal{C}$ is a *non-sensitive* attribute relevant to the downstream task (e.g., occupation). For example, an image depicting a male engineer may be annotated with $a_i = \texttt{male}$ and $c_i = \texttt{engineer}$.

Let $M$ be a VLM such as CLIP, and let $f_M$ denote its encoder that maps both text and images into a shared embedding space. Given a text query $Q$, define its embedding as $E_Q = f_M(Q)$; given an image $I$, define its embedding as $E_I = f_M(I)$. The alignment between $Q$ and $I$ is quantified by a distance measure $d(E_Q, E_I)$, which serves as the core similarity metric throughout downstream task like zero-shot classification and retrieval.

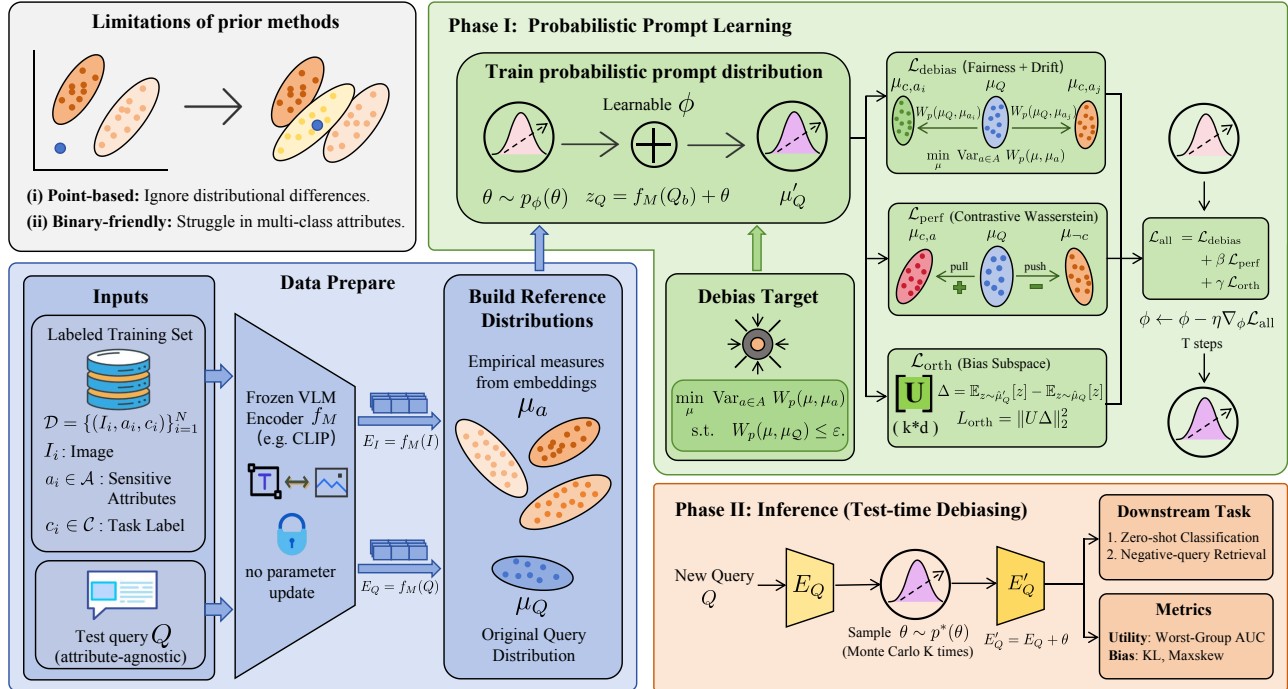

Figure 1. Overview of W4D. W4D performs test-time debiasing by constructing group reference distributions, learning probabilistic prompts, and applying them at inference to produce debiased query embeddings for downstream tasks.

## 3. Methodology

In this section, we first formalize VLM debiasing and define a distributional target over query embeddings. We then instantiate it with Wasserstein distance to handle multi-class attributes in a geometry-aware way. To preserve retrieval semantics during debiasing, we introduce a contrastive Wasserstein loss using concept-consistent positives and concept-inconsistent negatives. Finally, we implement the test-time optimization via lightweight probabilistic prompts.

### 3.1. VLM Debias Problem

We study debiasing for a VLM that retrieves images given an attribute-agnostic text query. Let $\mathcal{A}$ be the set of sensitive-attribute values and let $A \in \mathcal{A}$ denote the sensitive attribute associated with each image in the retrieval corpus.

**Retrieval-Induced Attribute Shift.** Let $\mathbb{P}_\mathcal{Q}$ be the distribution of such queries $Q$. Passing queries through the text encoder $f_M$ induces a distribution over text embeddings:

$$\mu_\mathcal{Q} := (f_M)_\# \mathbb{P}_\mathcal{Q} \in \mathcal{P}(\mathbb{R}^d). \quad (1)$$

Given a query $Q$, the VLM produces a ranking of images (e.g., by cosine similarity in the joint embedding space). Let $\mathcal{R}(Q)$ denote the retrieval operator that returns the top-$K$ images for $Q$. This retrieval procedure induces an *attribute distribution among retrieved images*:

$$\hat{\rho}_Q(a) := \mathbb{P}\big(A = a \,\big|\, I \in \mathcal{R}(Q)\big), \qquad a \in \mathcal{A}, \quad (2)$$

and, aggregated over the query distribution,

$$\hat{\rho}_\mathcal{Q}(a) := \mathbb{E}_{Q \sim \mathbb{P}_\mathcal{Q}}\big[\hat{\rho}_Q(a)\big]. \quad (3)$$

In many real-world settings, $\hat{\rho}_\mathcal{Q}$ can deviate substantially from the dataset attribute prior $\rho_0(a) := \mathbb{P}(A = a)$, even when queries are attribute-agnostic, due to spurious correlations in the learned embedding space between query semantics and sensitive attributes.

**Fairness objective.** Our goal is to mitigate this retrieval-induced shift by enforcing that the marginal attribute distribution among retrieved images matches the dataset prior:

$$\hat{\rho}_\mathcal{Q}(a) \approx \rho_0(a), \qquad \forall a \in \mathcal{A}. \quad (4)$$

**Debiasing target.** Let $\mu_{a\,a \in \mathcal{A}}$ be a family of sensitive-attribute reference image distributions in the joint embedding space, where each $\mu_a$ represents the empirical distribution of image embeddings associated with attribute value $a$ in the retrieval corpus. Intuitively, if the query-induced retrieval distribution is closer (under a suitable distributional distance) to some $\mu_a$ than to others, the retrieved results are more likely to over-represent images with attribute $a$. Hence, we aim to transform the original query-embedding distribution $\mu_\mathcal{Q}$ into a debiased distribution $\widetilde{\mu}_\mathcal{Q}$ that is *equidistant*

to all reference distributions under a generic distance $d(\cdot, \cdot)$ on $\mathcal{P}(\mathbb{R}^d)$:

$$d(\widetilde{\mu}_{\mathcal{Q}}, \mu_a) = d(\widetilde{\mu}_{\mathcal{Q}}, \mu_{a'}) \quad \text{for all } a, a' \in \mathcal{A}. \quad (5)$$

At the same time, the debiased distribution should preserve the semantics of the original queries; we enforce this by constraining $\widetilde{\mu}_{\mathcal{Q}}$ to remain close to $\mu_{\mathcal{Q}}$ under the same distance $d$. Putting these requirements together, we define debiasing target as:

$$\widetilde{\mu}_{\mathcal{Q}} \in \arg\min_{\mu \in \mathcal{P}(\mathbb{R}^d)} \quad \text{Var}_{a \in \mathcal{A}}\big(d(\mu, \mu_a)\big) \\ \text{s.t.} \quad d(\mu, \mu_{\mathcal{Q}}) \leq \varepsilon. \quad (6)$$

The variance objective encourages $\mu$ to be equally close to each $\mu_a$, which operationalizes the notion of attribute-agnosticity in distributional terms. The constraint acts as a soft regularization that limits semantic drift by keeping $\widetilde{\mu}_{\mathcal{Q}}$ within a radius-$\varepsilon$ ball around the original query-embedding distribution $\mu_{\mathcal{Q}}$ under $d$.

## 3.2. Debias in Distributional Setting

W4D is motivated by *two limitations* of previous test-time debiasing methods. **(i) Point-based debiasing:** many methods solve for a single ideal debiased embedding point, yet sensitive groups form structured clusters with distinct spread and geometry, so matching a point statistic can still leave sample-level disparities. To address this, we adopt a distributional view that represents each group as an embedding distribution and debiases by aligning distributions rather than a single point. We instantiate $d(\cdot, \cdot)$ with the geometry-aware $p$-Wasserstein distance $W_p$, which captures differences in both location and shape even when group supports overlap little. **(ii) Less support for multi-class attributes:** many objectives are tailored to *binary* attributes and lack a unified treatment of the multi-class setting. Under our distributional perspective, multi-class fairness naturally corresponds to making the debiased query distribution uniformly close to all group reference distributions $\mu_{a_{a \in \mathcal{A}}}$. Instead of imposing many pairwise constraints, we summarize cross-group disparity via the variance of group-wise Wasserstein distances, yielding a single permutation-invariant objective that scales gracefully with $|\mathcal{A}|$.

**Distributional problem formulation.** Recall that attribute-agnostic queries are drawn from $\mathbb{P}_{\mathcal{Q}}$ and the fixed text encoder induces $\mu_{\mathcal{Q}} = (f_M)_{\#}\mathbb{P}_{\mathcal{Q}} \in \mathcal{P}_p(\mathbb{R}^d)$. For each sensitive attribute $a \in \mathcal{A}$, we construct a reference embedding distribution $\mu_a \in \mathcal{P}_p(\mathbb{R}^d)$ (see next subsection for how $\mu_a$ is built). We seek a debiased distribution $\widetilde{\mu}_{\mathcal{Q}}$ that (i) is equally close to all $\mu_a$'s (fairness) and (ii) stays close to the original $\mu_{\mathcal{Q}}$:

$$\widetilde{\mu}_{\mathcal{Q}} \in \arg\min_{\mu \in \mathcal{P}_p(\mathbb{R}^d)} \quad \text{Var}_{a \in \mathcal{A}}\big(W_p(\mu, \mu_a)\big) \\ \text{s.t.} \quad W_p(\mu, \mu_{\mathcal{Q}}) \leq \varepsilon. \quad (7)$$

Here, $\text{Var}_{a \in \mathcal{A}}(\cdot)$ penalizes cross-group disparity in Wasserstein distances, implementing multi-class fairness in a single objective, while the Wasserstein-ball constraint prevents excessive semantic drift. Under the soft-assignment model $p(a \mid Q) \propto \rho_0(a)\exp(-\alpha W(\widetilde{\mu}_Q, \mu_a)^2)$, equalizing $W(\widetilde{\mu}_Q, \mu_a)$ across $a$ recovers $p(a \mid Q) \approx \rho_0$ (Appendix A).

In practice, we optimize an unconstrained surrogate by absorbing the constraint into the objective:

$$\mathcal{L}_{\text{debias}}(\mu) := \text{Var}_{a \in \mathcal{A}}\big(W_p(\mu, \mu_a)\big) + \lambda\big(W_p(\mu, \mu_{\mathcal{Q}}) - \varepsilon\big)_+^2, \quad (8)$$

where $(x)_+ := \max\{0, x\}$ and $\lambda > 0$ controls the strength of semantic preservation. We then obtain $\widetilde{\mu}_{\mathcal{Q}} \in \arg\min_{\mu} \mathcal{L}_{\text{debias}}(\mu)$.

## 3.3. Performance Preservation

While $\mathcal{L}_{\text{debias}}$ enforces *group-balanced alignment*, it does not by itself guarantee that the debiased query embeddings remain predictive of the intended retrieval concept. To explicitly preserve task semantics, we additionally regularize $\widetilde{\mu}_{\mathcal{Q}}$ to stay close to concept-consistent image embeddings and far from concept-inconsistent ones.

Fix a query concept $c$ (e.g., `engineer`). Let $\mu_{c,a} \in \mathcal{P}_p(\mathbb{R}^d)$ denote the image-embedding distribution of samples that satisfy concept $c$ *and* belong to sensitive group $a \in \mathcal{A}$ (positives), and let $\mu_{\neg c} \in \mathcal{P}_p(\mathbb{R}^d)$ denote the image-embedding distribution of samples that do *not* satisfy concept $c$ (negatives), aggregated over $a$. Our goal is to make $\widetilde{\mu}_{\mathcal{Q}}$ simultaneously (i) close to every $\mu_{c,a}$ to preserve the concept across groups, and (ii) separated from $\mu_{\neg c}$ to avoid drifting toward negatives.

**Wasserstein contrastive preservation loss.** We define a balanced, contrastive performance-preservation loss:

$$\mathcal{L}_{\text{perf}}(\mu) := \frac{1}{|\mathcal{A}|} \sum_{a \in \mathcal{A}} W_p(\mu, \mu_{c,a}) \\ + \alpha\left(m + \frac{1}{|\mathcal{A}|}\sum_{a \in \mathcal{A}} W_p(\mu, \mu_{c,a}) - W_p(\mu, \mu_{\neg c})\right)_+, \quad (9)$$

where $\alpha > 0$ weights the separation term and $m > 0$ is a margin. The first term draws $\mu$ toward $\mu_{c,a} a \in \mathcal{A}$ across groups, and the second enforces a margin $m$ against $\mu_{\neg c}$.

**Bias-Subspace Orthogonality Loss.** To prevent semantic-preserving updates from reintroducing sensitive-attribute information, we constrain the *distributional shift* from $\mu_{\mathcal{Q}}$ to $\mu$ to lie in the orthogonal complement of a sensitive subspace. Let $U \in \mathbb{R}^{k \times d}$ be an orthonormal basis of a "bias subspace" in the shared $d$-dimensional embedding space, where $k \ll d$ is the subspace dimension (typically $k =$

$|\mathcal{A}| - 1$) capturing sensitive directions. We define the mean-shift $\Delta(\mu) := \mathbb{E}_{z \sim \mu}[z] - \mathbb{E}_{z \sim \mu_{\mathcal{Q}}}[z] \in \mathbb{R}^d$. We penalize the projection of $\Delta(\mu)$ onto this subspace:

$$\mathcal{L}_{\text{orth}}(\mu) := \|U \, \Delta(\mu)\|_2^2. \qquad (10)$$

Intuitively, $\mathcal{L}_{\text{orth}}$ encourages performance-preserving updates orthogonal to sensitive directions, improving disentanglement between semantic alignment and debiasing.

**Overall objective.** Finally, we augment the debiasing objective with performance preservation by defining:

$$\mathcal{L}_{\text{all}}(\mu) := \mathcal{L}_{\text{debias}}(\mu) \; + \; \beta \, \mathcal{L}_{\text{perf}}(\mu) \; + \; \gamma \, \mathcal{L}_{\text{orth}}(\mu), \quad (11)$$

and solve $\min_{\mu \in \mathcal{P}_p(\mathbb{R}^d)} \mathcal{L}_{\text{all}}(\mu)$ where $\beta, \gamma > 0$ control the strength of semantic preservation and orthogonality.

### 3.4. Implementation of Test-Time Debiasing

Prompt learning offers a lightweight way to mitigate retrieval-induced attribute shift without updating the VLM (Ding et al., 2024; He et al., 2022; Lu et al., 2026; Xu et al., 2025a). By steering query embeddings, text-side prompts can counter spurious correlations in the shared space, aligning with our distributional Wasserstein objective. Modeling prompts as a perturbation distribution captures heterogeneous bias directions while limiting semantic drift.

In probabilistic prompts (Yang et al., 2023; Kwon et al., 2023; Yang et al., 2026a; Lu & Yin, 2025; Xu et al., 2025b), we treat the set of learnable prompts as a distribution over possible perturbations of the query embeddings. Instead of using a fixed set of prompts, we model the prompts as a continuous distribution $p(\theta)$, where $\theta$ represents a perturbation variable parameterized by learnable distribution parameters. Each query $Q$ is transformed by taking the expectation under this distribution, yielding an adjusted embedding:

$$E'_Q = f_M(Q) + \int \theta \, p(\theta) \, d\theta, \qquad (12)$$

where $f_M(Q)$ is the original query embedding, and $\int \theta \, p(\theta) \, d\theta$ is the expected prompt-induced shift.

This induces a *prompted* query-embedding distribution $\mu'_{\mathcal{Q}}$ (i.e., the distribution of $E'_Q$ when $Q \sim \mathbb{P}_{\mathcal{Q}}$). We then learn $p(\theta)$ by directly optimizing the same Wasserstein-based objective defined above, by substituting $\mu \leftarrow \mu'_{\mathcal{Q}}$ in the overall objective (debiasing with Wasserstein-distance variance, performance preservation via $\mathcal{L}_{\text{perf}}$, and orthogonality via $\mathcal{L}_{\text{orth}}$, with the Wasserstein-ball penalty around $\mu_{\mathcal{Q}}$). Concretely,

$$p^*(\theta) \; = \; \arg\min_{p(\theta)} \mathcal{L}_{\text{all}}(\mu'_{\mathcal{Q}}), \qquad (13)$$

where $\mathcal{L}_{\text{all}}(\cdot)$ is computed according to (11).

At test time, for a new query $Q$, we apply the learned distribution $p^*(\theta)$ to obtain:

$$E'_Q = f_M(Q) + \int \theta \, p^*(\theta) \, d\theta, \qquad (14)$$

thereby achieving test-time debiasing via probabilistic prompts, without modifying the underlying VLM.

## 4. Experiments

### 4.1. Experiment Results

We evaluate on four face image datasets: Fair-Face (Karkkainen & Joo, 2021), CelebA (Liu et al., 2015), UTKFace (Zhang et al., 2017), and Facet (Gustafson et al., 2023). CelebA includes gender annotations only; FairFace and UTKFace provide gender and race, while Facet annotates gender and skin tone.

**Models.** To demonstrate the generality of our method, we conduct experiments on two widely used CLIP variants (Radford et al., 2021), namely CLIP-ViT-Base-Patch16 and CLIP-ViT-Large-Patch14.

**Baselines.** We compare our method against the following representative debiasing baselines.

- **Baseline CLIP** directly uses the original CLIP model without any debiasing or adaptation.

- **DeAR** (Seth et al., 2023) learns an additive residual to adjust VLM image embeddings, suppressing identity-related spurious correlations with minimal perturbation.

- **Orthogonal Projection (Orth-Proj.)** (Chuang et al., 2023) removes bias by projecting query embeddings to be orthogonal to a global spurious-attribute subspace, eliminating sensitive-attribute directions.

- **Orthogonal Calibration (Orth-Cal.)** (Chuang et al., 2023) enforces orthogonality to the spurious-attribute subspace and adds a calibration regularizer to align attribute-augmented query variants after projection.

- **BendVLM** (Gerych et al., 2024) performs test-time debiasing via a query-adaptive nonlinear projection guided by a sensitive-attribute prior, while preserving performance.

- **SFID** (Jung et al., 2024) reduces multimodal bias via feature pruning and low-confidence filling, limiting spurious-attribute reliance while preserving semantics.

**Tasks and Evaluation Metrics.** We evaluate both *utility* and *bias* under two query-driven settings. Given a text query $Q$, we rank images by similarity $s(Q, I) = 1 - d(f_M(Q), f_M(I))$ and consider the top-$m$ retrieved set $\mathcal{R}_m(Q)$.

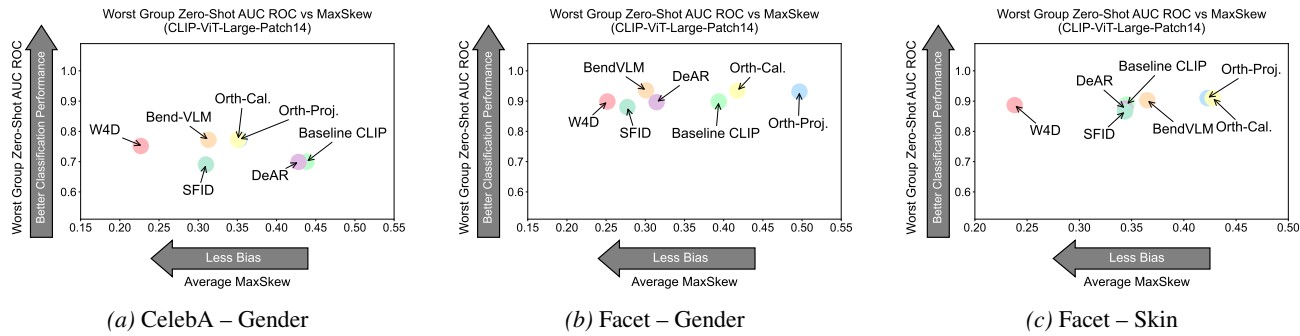

*Figure 2.* Comparison of W4D and prior test-time debiasing methods on CLIP-ViT-Large-Patch14.

*Table 1.* Debiasing UTKFACE and FAIRFACE dataset with respect to `gender` and `race` for STEREOTYPE queries.

| Attribute | Method | UTKFACE | | | | FAIRFACE | | | |
| | | CLIP-ViT-B-P16 | | CLIP-ViT-L-P14 | | CLIP-ViT-B-P16 | | CLIP-ViT-L-P14 | |
| | | KL Div.↓ | MaxSkew↓ | KL Div.↓ | MaxSkew↓ | KL Div.↓ | MaxSkew↓ | KL Div.↓ | MaxSkew↓ |
|---|---|---|---|---|---|---|---|---|---|
| Gender | Baseline CLIP | $0.1575 \pm 0.0131$ | $0.8204 \pm 0.0571$ | $0.0738 \pm 0.0065$ | $0.4757 \pm 0.0290$ | $0.1179 \pm 0.0101$ | $0.6662 \pm 0.0444$ | $0.0813 \pm 0.0079$ | $0.5011 \pm 0.0377$ |
| | Orth-Proj. | $0.1470 \pm 0.0036$ | $0.7092 \pm 0.0148$ | $0.0351 \pm 0.0074$ | $0.2981 \pm 0.0413$ | $0.3155 \pm 0.0115$ | $1.3983 \pm 0.0612$ | $0.0257 \pm 0.0014$ | $0.2005 \pm 0.0150$ |
| | Orth-Cal. | $0.2048 \pm 0.0040$ | $0.9348 \pm 0.0184$ | $0.0181 \pm 0.0033$ | $0.1776 \pm 0.0267$ | $0.4043 \pm 0.0106$ | $1.8997 \pm 0.0523$ | $0.0373 \pm 0.0046$ | $0.2422 \pm 0.0159$ |
| | BendVLM | $\underline{0.0096 \pm 0.0022}$ | $\underline{0.1248 \pm 0.0147}$ | $0.0182 \pm 0.0054$ | $0.1756 \pm 0.0325$ | $\underline{0.0267 \pm 0.0030}$ | $\underline{0.2063 \pm 0.0068}$ | $\mathbf{0.0057 \pm 0.0013}$ | $\mathbf{0.0952 \pm 0.0166}$ |
| | DeAR | $0.1315 \pm 0.0629$ | $0.7020 \pm 0.2855$ | $0.0735 \pm 0.0057$ | $0.4745 \pm 0.0252$ | $0.0861 \pm 0.0109$ | $0.4853 \pm 0.0618$ | $0.0667 \pm 0.0221$ | $0.4234 \pm 0.1168$ |
| | SFID | $0.0391 \pm 0.0102$ | $0.3074 \pm 0.0555$ | $\underline{0.0039 \pm 0.0014}$ | $\underline{0.0790 \pm 0.0186}$ | $0.0548 \pm 0.0123$ | $0.3729 \pm 0.0587$ | $0.0368 \pm 0.0087$ | $0.2547 \pm 0.0403$ |
| | W4D | $\mathbf{0.0048 \pm 0.0038}$ | $\mathbf{0.0833 \pm 0.0401}$ | $\mathbf{0.0037 \pm 0.0042}$ | $\mathbf{0.0674 \pm 0.0479}$ | $\mathbf{0.0150 \pm 0.0091}$ | $\mathbf{0.1614 \pm 0.0702}$ | $\underline{0.0097 \pm 0.0072}$ | $\underline{0.1324 \pm 0.0712}$ |
| Race | Baseline CLIP | $0.1227 \pm 0.0059$ | $1.0074 \pm 0.0615$ | $0.0984 \pm 0.0096$ | $0.8773 \pm 0.0655$ | $0.1882 \pm 0.0067$ | $1.3405 \pm 0.0625$ | $0.1865 \pm 0.0120$ | $1.2872 \pm 0.0733$ |
| | Orth-Proj. | $0.1794 \pm 0.0258$ | $1.4795 \pm 0.2020$ | $0.1246 \pm 0.0132$ | $1.3198 \pm 0.1190$ | $0.2624 \pm 0.0240$ | $1.4336 \pm 0.0807$ | $0.1617 \pm 0.0092$ | $1.1865 \pm 0.0438$ |
| | Orth-Cal. | $0.1739 \pm 0.0258$ | $1.3721 \pm 0.1885$ | $0.1150 \pm 0.0122$ | $1.1798 \pm 0.1081$ | $0.2558 \pm 0.0227$ | $1.4099 \pm 0.0541$ | $0.1582 \pm 0.0094$ | $1.2596 \pm 0.0592$ |
| | BendVLM | $\underline{0.0724 \pm 0.0094}$ | $\underline{0.7140 \pm 0.0794}$ | $0.0629 \pm 0.0154$ | $0.6207 \pm 0.0546$ | $\underline{0.0879 \pm 0.0091}$ | $\underline{0.8586 \pm 0.1161}$ | $0.0946 \pm 0.0050$ | $0.9714 \pm 0.0501$ |
| | DeAR | $0.1160 \pm 0.0093$ | $0.9568 \pm 0.0599$ | $0.0931 \pm 0.0187$ | $0.8563 \pm 0.0916$ | $0.1738 \pm 0.0257$ | $1.2641 \pm 0.1593$ | $0.1780 \pm 0.0267$ | $1.2581 \pm 0.1199$ |
| | SFID | $0.0838 \pm 0.0072$ | $0.8302 \pm 0.0435$ | $\underline{0.0387 \pm 0.0110}$ | $\underline{0.5419 \pm 0.0991}$ | $0.1360 \pm 0.0048$ | $1.0626 \pm 0.0589$ | $0.1420 \pm 0.0062$ | $1.1098 \pm 0.0334$ |
| | W4D | $\mathbf{0.0201 \pm 0.0079}$ | $\mathbf{0.3312 \pm 0.1075}$ | $\mathbf{0.0218 \pm 0.0008}$ | $\mathbf{0.3133 \pm 0.0378}$ | $\mathbf{0.0383 \pm 0.0080}$ | $\mathbf{0.5083 \pm 0.0881}$ | $\mathbf{0.0466 \pm 0.0113}$ | $\mathbf{0.6639 \pm 0.1178}$ |

**(1) Zero-shot classification.** We perform zero-shot classification on non-sensitive labels $c \in \mathcal{C}$ using a query set $\mathcal{Q}_{cls}$ (e.g., attribute-agnostic templates describing classes). We report **Worst-Group AUC-ROC** as our utility metric.

To quantify **retrieval bias**, we compare the sensitive-attribute composition of retrieved results against the dataset prior. Let $P_a$ denote the marginal distribution of the sensitive attribute $a$ in the target dataset. For each query $Q$, define the empirical attribute distribution among the top-$m$ retrieved images $\mathcal{R}_m(Q)$ as

$$\hat{P}_a(Q)(a) := \frac{1}{|\mathcal{R}_m(Q)|} \sum_{I_i \in \mathcal{R}_m(Q)} \mathbf{1}[a_i = a], \quad (15)$$

and average across queries,

$$\hat{P}_a := \frac{1}{|\mathcal{Q}_{cls}|} \sum_{Q \in \mathcal{Q}_{cls}} \hat{P}_a(Q). \quad (16)$$

Intuitively, if retrieval is insensitive to the spurious attribute, then $\hat{P}_a$ should match $P_a$. We report **KL divergence** $\mathrm{KL}[\hat{P}_a \parallel P_a]$ and **MaxSkew** as bias metrics, where smaller values indicate less retrieval bias. The exact computation of all metrics is provided in Appendix D.1.

**(2) Negative-query retrieval (STEREOTYPES).** We additionally evaluate a retrieval-only setting using a set of neg-

ative stereotype queries $\mathcal{Q}_{st}$ (denoted STEREOTYPES). Since these queries are designed to probe harmful spurious correlations, we focus *only* on bias: we compute $\hat{P}_a$ by aggregating $\hat{P}_a(Q)$ over $Q \in \mathcal{Q}_{st}$ and report $\mathrm{KL}[\hat{P}_a \parallel P_a]$ and MAXSKEW as above.

### 4.2. Main Results

We evaluate our method under two query-driven settings: (i) **zero-shot classification** on non-sensitive labels, where utility is measured by *Worst-Group AUC-ROC* and bias is measured by *KL divergence* and *MaxSkew*; and (ii) **negative-query retrieval** using stereotype/attacking queries, where we focus on bias metrics computed from the sensitive-attribute composition of the top-$m$ retrieved set.

**Zero-shot classification.** Across datasets and CLIP backbones, our method achieves a consistently better trade-off between classification utility and retrieval fairness. Concretely, we *preserve* Worst-Group AUC-ROC to be close to the base CLIP, while *reducing* retrieval bias more effectively than prior test-time baselines. This pattern is summarized by the AUC–MaxSkew scatterplots: our method tends to move toward the upper-left region, indicating that we can debias without sacrificing the base model's predictive signal. More detailed experimental results are provided in Appendix C.

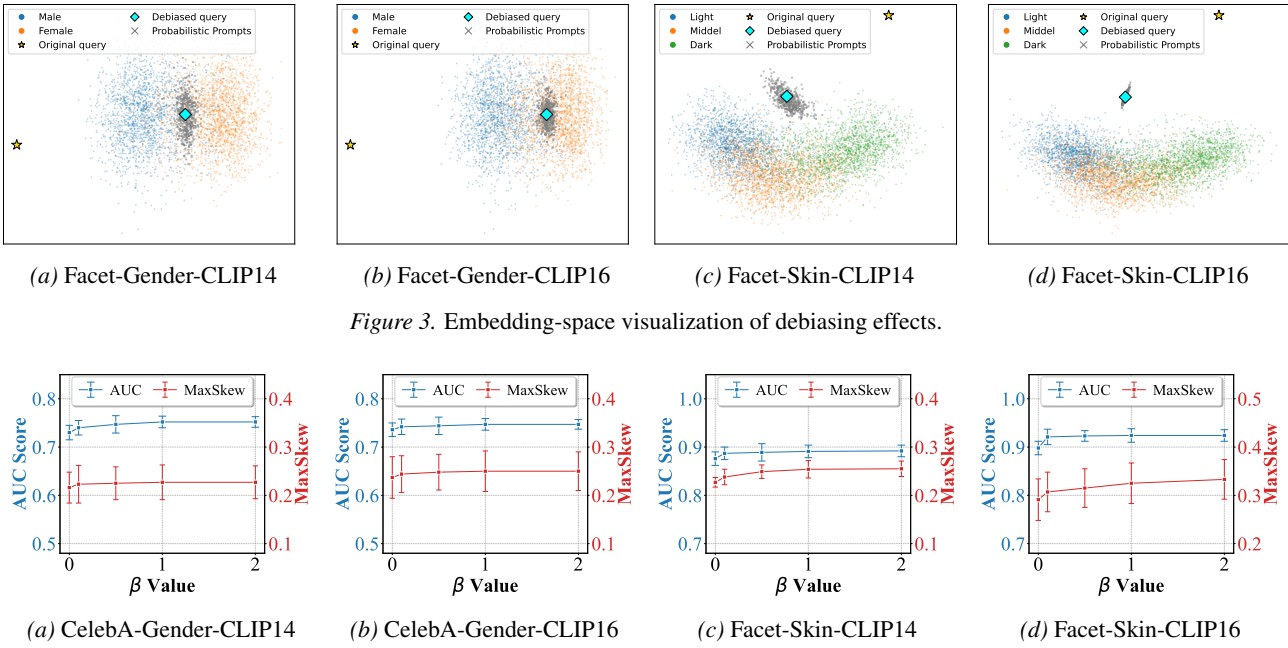

*Figure 3.* Embedding-space visualization of debiasing effects.

*(a)* CelebA-Gender-CLIP14  *(b)* CelebA-Gender-CLIP16  *(c)* Facet-Skin-CLIP14  *(d)* Facet-Skin-CLIP16

*Figure 4.* Effect of $\beta$ on debiasing and performance.

**Negative-query retrieval.** For stereotype queries, we observe strong debiasing improvements on both **binary** and **multi-class** sensitive attributes. Notably, the advantage is *larger* in the multi-class case: our method yields substantially lower KL/MaxSkew than competing approaches when the sensitive attribute has more than two categories, which matches our motivation of explicitly targeting multi-class fairness via distributional alignment.

### 4.3. Analysis

**(1) Visualization in the embedding space.** As shown in Figure 3, we provide a qualitative visualization to support our distributional claim: compared with the base model, the debiased query embeddings produced by our method exhibit *more similar distances* to the reference embedding distributions of different sensitive groups. In other words, group-wise clusters become less separated under our transformation, consistent with the goal of being equidistant to sensitive-group distributions.

**(2) Effect of $\beta$ (performance–debias trade-off).** We sweep $\beta \in \{0, 0.1, 0.5, 1.0, 2.0\}$ to study how strongly the objective emphasizes semantic preservation. Shown in Figure 4, as $\beta$ increases, the model prioritizes concept consistency more aggressively: **Worst-Group AUC-ROC increases**, but **MaxSkew also increases**. This indicates a clear trade-off: pushing too hard on performance preservation can re-amplify spurious reliance on the sensitive attribute.

**(3) Effect of $\gamma$ (orthogonality strength).** We sweep $\gamma \in \{0, 0.5, 1.0, 2.0, 3.0\}$ and observe the opposite trend. In Figure 5, as $\gamma$ increases, the update direction is more strongly constrained to avoid the sensitive subspace: **MaxSkew decreases**, while **Worst-Group AUC-ROC decreases**. This is expected: stronger orthogonality removes more sensitive-attribute signal but can also remove task-relevant directions that partially overlap with the bias subspace.

**(4) Sampling size $K$ for probabilistic prompts.** Our learned prompt is a *distribution*, so inference depends on Monte Carlo sampling. In Figure 6, we sweep $K \in \{16, 32, 64, 128, 256\}$. Increasing $K$ yields **better overall performance** and **smaller variance** across runs, suggesting that a larger sample budget provides a more reliable estimate of the prompt-induced shift and reduces stochasticity in retrieval outcomes.

## 5. Related Works

**Bias in VLMs.** A substantial line of work has established that VLMs encode social biases in their shared embedding spaces, reflecting stereotypical or imbalanced pretraining signals (Radford et al., 2021; Agarwal et al., 2021). For *gender*, even ostensibly neutral prompts can elicit systematically gendered associations in CLIP-style embeddings, suggesting that gender cues are implicitly entangled with semantic concepts rather than triggered only by explicit attribute words (Dehouche, 2021). Such biases are visible in open-vocabulary recognition, where performance gaps vary by class and aggregate into measurable demographic dispar-

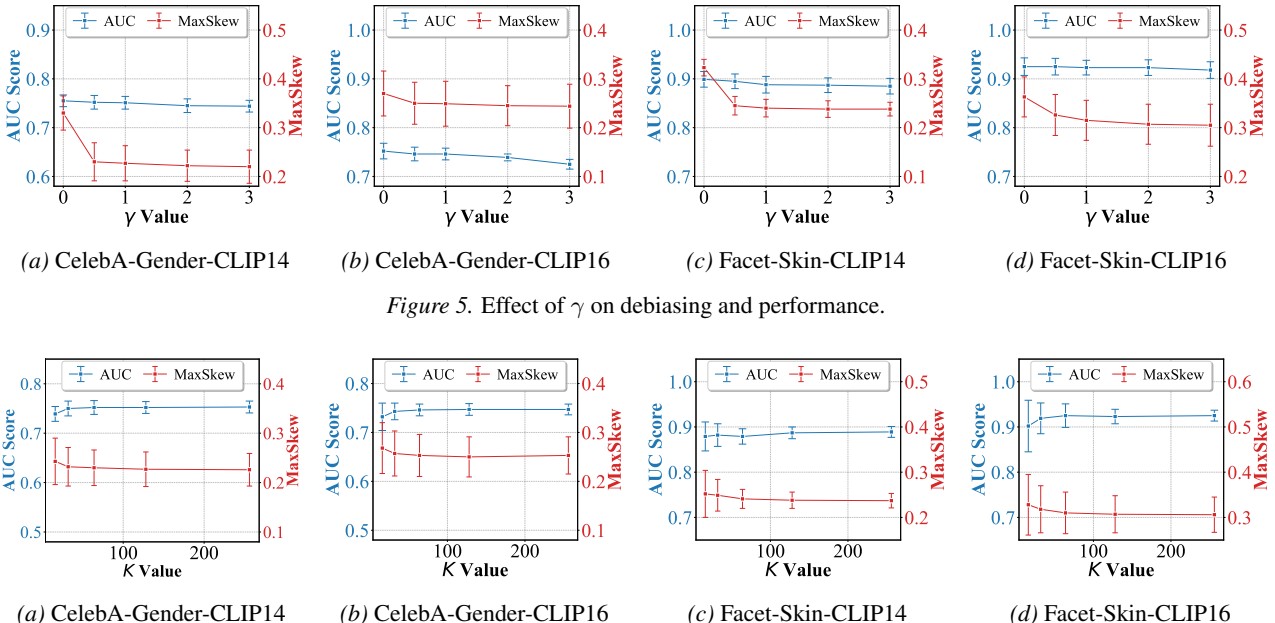

*Figure 5.* Effect of $\gamma$ on debiasing and performance.

*(a)* CelebA-Gender-CLIP14    *(b)* CelebA-Gender-CLIP16    *(c)* Facet-Skin-CLIP14    *(d)* Facet-Skin-CLIP16

*(a)* CelebA-Gender-CLIP14    *(b)* CelebA-Gender-CLIP16    *(c)* Facet-Skin-CLIP14    *(d)* Facet-Skin-CLIP16

*Figure 6.* Effect of Monte Carlo sample size $K$ on debiasing and performance.

ity; FACET operationalizes these effects in multi-class settings (Gustafson et al., 2023). For *race*, embeddings exhibit asymmetric alignment to racial descriptors and harmful mis-associations, indicating socially loaded correlations beyond task-relevant semantics (Barlas et al., 2021; Wolfe et al., 2022; Yang et al., 2026b). These representational skews translate into downstream harms: zero-shot classification inherits group gaps (Hall et al., 2023), and text-to-image retrieval can produce demographically skewed top-$K$ results even for neutral queries (Kong et al., 2023), amplifying representational imbalances and reinforcing stereotypes.

**Debiasing VLMs.** To mitigate social biases in VLMs, existing methods broadly fall into *training-based* approaches that update model parameters and *test-time* approaches that debias fixed backbones at inference. Training-based debiasing intervenes during learning, using adversarial prompt objectives to suppress protected-attribute cues (Berg et al., 2022), text-only optimization to mitigate bias without updating the visual encoder (Yang et al., 2024), or prompt-regularized debiased fine-tuning (Zhu et al., 2023). Complementarily, data-centric counterfactual training reduces spurious attribute–concept correlations by balancing data (Al-abdulmohsin et al., 2024). In contrast, *test-time* debiasing targets deployment settings where retraining or finetuning is impractical. A common paradigm is to elicit sensitive directions with biased prompts and suppress them via post-hoc projection in the embedding space (Chuang et al., 2023; Howard et al., 2024; Li et al., 2025). Beyond global subspace removal, DeAR (Seth et al., 2023) applies lightweight residual edits to repair biased cross-modal correspondences

while keeping encoders fixed. Other prompt-only calibration or reweighting strategies improve fairness without any parameter updates (Jang et al., 2025). Finally, more expressive inference-time adaptations move past a single linear bias direction: BendVLM (Gerych et al., 2024) performs input-adaptive nonlinear corrections, and SFID (Jung et al., 2024) suppresses attribute-correlated cues by selectively pruning and imputing uncertain features across modalities.

## 6. Conclusion

In this paper, we first identify two key limitations for test-time VLM debiasing, i.e., *insufficiency of point-based debiasing* and *limited support for multi-class attributes*. To address these limitations, we present W4D, a lightweight test-time debiasing framework that reconceptualizes fairness as a *distributional* alignment problem under the Wasserstein distance. This approach moves beyond point-wise corrections, thereby addressing limitation 1. By optimizing a variance-of-distances objective, W4D naturally scales to multi-class attributes while preserving semantic fidelity via tailored regularizers, addressing limitation 2. We further demonstrate that probabilistic prompts provide a practical mechanism to realize distributional perturbations at inference without updating any model parameters. Extensive experiments across multiple datasets show that W4D yields a stronger fairness–utility trade-off compared to existing methods. This distributional perspective opens a promising direction for VLM debiasing, particularly as it facilitates effective handling of multi-class attributes and operates purely at test-time, enhancing its practical applicability.

## Acknowledgements

This work was supported in part by the National Natural Science Foundation of China (No. 72192823 and 62402148).

## Impact Statement

This paper presents work whose goal is to advance the field of Machine Learning. There are many potential societal consequences of our work, none which we feel must be specifically highlighted here.

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

# A. Distance Equalization and Prior Matching

This appendix clarifies when *distance equalization* implies *prior matching* under a soft-assignment surrogate, and what modification is needed to exactly encode a *non-uniform* target prior $\rho_0$ via distances.

**Notation.** Let $\{\mu_a\}_{a\in\mathcal{A}}$ be sensitive-group reference distributions in the shared embedding space, and let $\tilde{\mu}_Q$ denote the (debiased) query-embedding distribution for a query $Q$. Write $d_a := W(\tilde{\mu}_Q, \mu_a)^2$ and let $\alpha > 0$ be a temperature.

## A.1. Model A: prior-weighted soft assignment

A common propensity surrogate multiplies a base prior $\rho_0$ with a distance-based score:

$$p_{\mathrm{A}}(a \mid Q) := \frac{\rho_0(a)\exp(-\alpha d_a)}{\sum_{a'\in\mathcal{A}}\rho_0(a')\exp(-\alpha d_{a'})}. \tag{17}$$

**Proposition A.1** (Exact prior matching under Model A). *For any target prior $\rho_0$ (uniform or not), the following are equivalent:*

$$p_{\mathrm{A}}(a \mid Q) = \rho_0(a) \ \ \forall a \in \mathcal{A} \quad \Longleftrightarrow \quad d_a = C \ \ \forall a \in \mathcal{A} \tag{18}$$

*for some constant $C$ independent of $a$.*

*Proof.* From (17),

$$\frac{p_{\mathrm{A}}(a \mid Q)}{\rho_0(a)} = \frac{\exp(-\alpha d_a)}{\sum_{a'}\rho_0(a')\exp(-\alpha d_{a'})}.$$

Hence $p_{\mathrm{A}}(a \mid Q) = \rho_0(a)$ for all $a$ holds iff $\exp(-\alpha d_a)$ is constant over $a$, i.e., $d_a = C$ for all $a$. The converse is immediate by substitution. $\square$

**Implication.** Proposition A.1 shows that under Model A, *distance equalization is sufficient (and necessary) to preserve the target prior $\rho_0$*, but *non-uniform $\rho_0$* cannot be enforced by making distances non-uniform: exact prior matching requires all $d_a$ to be equal regardless of $\rho_0$.

## A.2. Model B: distance-only soft assignment

If one instead defines propensity purely via distances,

$$p_{\mathrm{B}}(a \mid Q) := \frac{\exp(-\alpha d_a)}{\sum_{a'\in\mathcal{A}}\exp(-\alpha d_{a'})}, \tag{19}$$

then a non-uniform target prior can be *exactly* encoded by a prescribed distance profile.

**Proposition A.2** (Encoding a non-uniform prior via distances under Model B). *Let $\rho_0$ be any distribution on $\mathcal{A}$ with $\rho_0(a) > 0$ for all $a$. If*

$$d_a = C - \frac{1}{\alpha}\log\rho_0(a) \quad \forall a \in \mathcal{A}, \tag{20}$$

*then $p_{\mathrm{B}}(a \mid Q) = \rho_0(a)$ for all $a \in \mathcal{A}$.*

*Proof.* Substitute (20) into (19):

$$\exp(-\alpha d_a) = \exp\Big(-\alpha C + \log\rho_0(a)\Big) = \exp(-\alpha C)\,\rho_0(a).$$

Therefore,

$$p_{\mathrm{B}}(a \mid Q) = \frac{\exp(-\alpha C)\rho_0(a)}{\sum_{a'}\exp(-\alpha C)\rho_0(a')} = \frac{\rho_0(a)}{\sum_{a'}\rho_0(a')} = \rho_0(a),$$

since $\sum_{a'}\rho_0(a') = 1$. $\square$

**Connection to a variance objective.** Condition (20) is equivalent to requiring $d_a + \frac{1}{\alpha} \log \rho_0(a)$ to be constant over $a$. Thus, minimizing

$$\text{Var}_{a \sim \rho_0} \left[ d_a + \frac{1}{\alpha} \log \rho_0(a) \right] \tag{21}$$

encourages approximate prior matching under Model B when exact equality is infeasible.

**Takeaway.** Model A implies "equal distances $\Rightarrow$ preserve $\rho_0$" (for any $\rho_0$), whereas Model B allows "shaped distances $\Rightarrow$ realize non-uniform $\rho_0$" exactly.

# B. Practical Implementation

### B.1. Algorithm Procedure

**Entropy-regularized OT solver details.** All Wasserstein terms in Section 3 are instantiated with the 2-Wasserstein distance and computed between empirical measures using the *entropy-regularized* Kantorovich formulation solved by the Sinkhorn algorithm. Concretely, for two empirical distributions $\hat{\mu} = \frac{1}{n} \sum_{i=1}^{n} \delta_{x_i}$ and $\hat{\nu} = \frac{1}{m} \sum_{j=1}^{m} \delta_{y_j}$, we minimize

$$W_{2,\tau}^2(\hat{\mu}, \hat{\nu}) = \min_{\pi \in \Pi(\hat{\mu},\hat{\nu})} \langle C, \pi \rangle + \tau \, \mathrm{KL}(\pi \,\|\, \hat{\mu} \otimes \hat{\nu}), \tag{22}$$

where the ground cost is $C_{ij} = \|x_i - y_j\|_2^2$ in the shared embedding space. We run $L$ Sinkhorn iterations (in log-domain for numerical stability) to obtain the optimal coupling $\pi^\star = \mathrm{diag}(u) \exp(-C/\tau) \mathrm{diag}(v)$ and use $\sqrt{\langle C, \pi^\star \rangle}$ as the corresponding regularized $W_2$ estimate. In practice, each OT call is evaluated on mini-batches by subsampling support points from each empirical measure, yielding per-call complexity $O(L\,nm)$ and enabling scalable training/inference within Algorithm 1.

In practice, we represent $p_\phi(\theta)$ by a finite set of $M$ learnable prompt offsets $\{\theta_m\}_{m=1}^M$, i.e., an empirical distribution

$$\hat{p}_\phi = \frac{1}{M} \sum_{m=1}^{M} \delta_{\theta_m}. \tag{23}$$

During inference, we perform Monte Carlo estimation by sampling $K$ offsets from $\hat{p}_\phi$ (with replacement) to approximate the expectations in our objective. Additionally, for semantic preservation, we use a simple cosine-similarity regularizer between the mean prompted embedding and the original query embedding.

---

**Algorithm 1** Test-time Debiasing via Probabilistic Prompts

---

1: **Inputs:** training set $\mathcal{D} = \{(I_i, a_i, c_i)\}_{i=1}^N$
2:     frozen encoder $f_M$; sensitive attributes $\mathcal{A}$
3:     concept $c$; bias basis $U$; loss weights $(\beta, \gamma)$;
4:     learning rate $\eta$; steps $T$
5: **Init:** prompt distribution $p_\phi$
6: **Build reference measures.**
7: **for all** $a \in \mathcal{A}$ **do**
8:     $\hat{\mu}_a \leftarrow \frac{1}{|\mathcal{D}_a|} \sum_{i:a_i=a} \delta_{f_M(I_i)}$
9:     $\hat{\mu}_{c,a} \leftarrow \frac{1}{|\mathcal{D}_{c,a}|} \sum_{i:c_i=c,a_i=a} \delta_{f_M(I_i)}$
10: **end for**
11: $\hat{\mu}_{\neg c} \leftarrow \frac{1}{|\mathcal{D}_{\neg c}|} \sum_{i:c_i \neq c} \delta_{f_M(I_i)}$
12: $\hat{\mu}_{\mathcal{Q}} \leftarrow \frac{1}{|Q_b|} \sum_{Q \in Q_b} \delta_{f_M(Q)}$
13: **Phase I: Optimize prompt tensor.**
14: **for** $t = 1$ **to** $T$ **do**
15:     Sample queries $Q_b \sim \mathbb{P}_{\mathcal{Q}}$
16:     $z_Q \leftarrow f_M(Q_b) + p_\phi$
17:     $\hat{\mu}'_{\mathcal{Q}} \leftarrow \frac{1}{|Q_b|} \sum_{Q \in Q_b} \delta_{z_Q}$
18:     Compute $\mathcal{L}_{\mathrm{debias}}(\hat{\mu}'_{\mathcal{Q}})$ using Eq. (8)
19:     Compute $\mathcal{L}_{\mathrm{perf}}(\hat{\mu}'_{\mathcal{Q}})$ using Eq. (9)
20:     Compute $\mathcal{L}_{\mathrm{orth}}(\hat{\mu}'_{\mathcal{Q}})$ using Eq. (10)
21:     $\mathcal{L} \leftarrow \mathcal{L}_{\mathrm{debias}} + \beta \mathcal{L}_{\mathrm{perf}} + \gamma \mathcal{L}_{\mathrm{orth}}$
22:     $\phi \leftarrow \phi - \eta \nabla_\phi \mathcal{L}$
23: **end for**
24: Set $p^* \leftarrow p_\phi$
25: **Phase II: Inference.**
26: $E_Q \leftarrow f_M(Q)$; sample $\theta \sim p^*(\theta)$
27: **return** $E'_Q \leftarrow E_Q + \theta$

---

## B.2. Empirical Computation.

In practice, the population embedding distributions $\{\nu_a\}_{a \in \mathcal{A}}$ and $\{\nu_c\}_{c \in \mathcal{C}}$ are unknown, hence directly optimizing over them is infeasible. Given the labeled training set $\mathcal{D} = \{(I_i, a_i, c_i)\}_{i=1}^N$, we construct empirical measures in the embedding space:

$$\hat{\nu}_a := \frac{1}{|\mathcal{D}_a|} \sum_{i:\, a_i = a} \delta_{f_M(I_i)}, \quad \hat{\nu}_c := \frac{1}{|\mathcal{D}_c|} \sum_{i:\, c_i = c} \delta_{f_M(I_i)}. \tag{24}$$

Following previous studies (Jiang et al., 2020; Klenke, 2008; Zeng et al., 2026; Zheng et al., 2022; Du et al., 2024; Xu et al., 2026; Li et al., 2021), when $|\mathcal{D}_a|$ (and $|\mathcal{D}_c|$) is large, these empirical measures approximate the corresponding population measures in Wasserstein distance, i.e., $W_p(\hat{\nu}_a, \nu_a) \to 0$ and $W_p(\hat{\nu}_c, \nu_c) \to 0$ as sample size grows. Moreover, by the triangle inequality, empirical Wasserstein distances are close to their population counterparts; for any distributions $(\mu, \nu)$ and their empirical versions $(\hat{\mu}, \hat{\nu})$,

$$\left| W_p(\hat{\mu}, \hat{\nu}) - W_p(\mu, \nu) \right| \leq W_p(\hat{\mu}, \mu) + W_p(\hat{\nu}, \nu), \tag{25}$$

so the discrepancy vanishes as the empirical measures converge. Therefore, computing Wasserstein terms using $\hat{\nu}_a, \hat{\nu}_c$ provides a consistent approximation to the population-level objective.

## B.3. Complexity Analysis.

Let $d$ be the embedding dimension, $|\mathcal{A}|$ the number of sensitive groups, $B := |Q_b|$ the query minibatch size, and $T$ the number of training steps. Denote by $C_M(Q_b)$ the cost of encoding $B$ queries with the frozen text encoder. Under the Sinkhorn OT approximation with $L$ iterations, computing $W_p$ between two empirical measures with $n$ and $m$ support points costs $C_W(n, m) = O(Lnm)$. In each step, we compute Wasserstein distances from $\hat{\mu}'_Q$ (with $B$ support points) to $\{\hat{\mu}_a\}_{a \in \mathcal{A}}$, $\hat{\mu}_Q$, $\{\hat{\mu}_{c,a}\}_{a \in \mathcal{A}}$, and $\hat{\mu}_{\neg c}$. Let $m_{\text{ref}}$ denote the (typical) number of support points in these reference measures; then the OT cost per step is $O(|\mathcal{A}| C_W(B, m_{\text{ref}})) = O(|\mathcal{A}| L B m_{\text{ref}})$ up to constant factors. The orthogonality term requires only minibatch means and a $k \times d$ projection, costing $O(Bd + kd)$. Therefore, the total training time is

$$O\Big( T\big(C_M(Q_b) + |\mathcal{A}| L B m_{\text{ref}} + Bd + kd\big)\Big), \tag{26}$$

and inference for a single query costs $O\big(C_M(Q) + Kd\big)$, where $K$ is the number of Monte Carlo samples used at test time.

# C. Additional Experiments

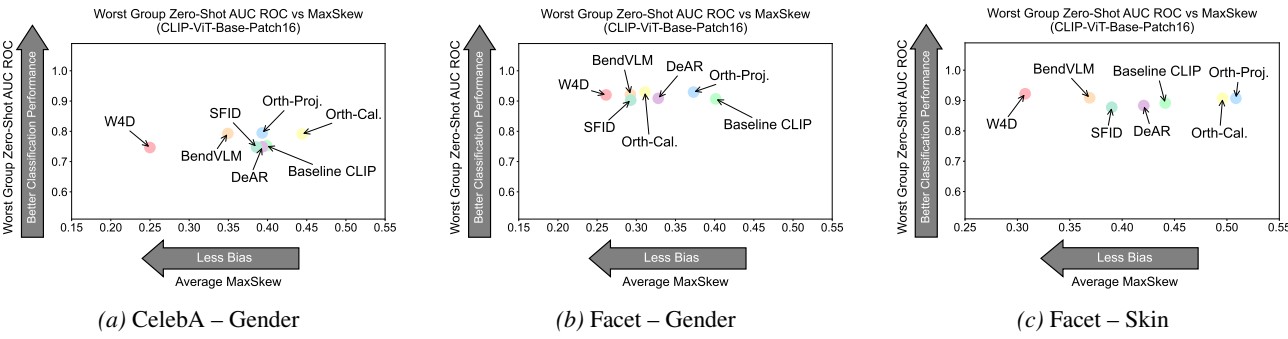

*(a)* CelebA – Gender  *(b)* Facet – Gender  *(c)* Facet – Skin

*Figure 7.* Comparison of W4D and prior test-time debiasing methods on CLIP-ViT-Base-Patch16.

*Table 2.* Debiasing the CELEBA dataset with respect to `gender` for HAIR COLOR queries.

| | | CLIP-ViT-B-P16 | | | CLIP-ViT-L-P14 | | |
|---|---|---|---|---|---|---|---|
| **Attribute** | **Method** | **KL Div.↓** | **MaxSkew↓** | **Worst Group AUC↑** | **KL Div.↓** | **MaxSkew↓** | **Worst Group AUC↑** |
| | Baseline CLIP | $0.0739 \pm 0.0062$ | $0.3988 \pm 0.0570$ | $0.7525 \pm 0.0138$ | $0.1075 \pm 0.0163$ | $0.4382 \pm 0.0389$ | $0.6995 \pm 0.0120$ |
| | Orth-Proj. | $0.0970 \pm 0.0102$ | $0.3932 \pm 0.0327$ | $\mathbf{0.7947 \pm 0.0124}$ | $0.0690 \pm 0.0092$ | $0.3526 \pm 0.0239$ | $\mathbf{0.7734 \pm 0.0068}$ |
| | Orth-Cal. | $0.0655 \pm 0.0040$ | $0.4440 \pm 0.0430$ | $0.7910 \pm 0.0125$ | $0.0537 \pm 0.0076$ | $0.3510 \pm 0.0169$ | $\underline{0.7729 \pm 0.0078}$ |
| Gender | BendVLM | $\underline{0.0399 \pm 0.0085}$ | $\underline{0.3493 \pm 0.0236}$ | $0.7928 \pm 0.0130$ | $\underline{0.0407 \pm 0.0116}$ | $0.3132 \pm 0.0384$ | $0.7721 \pm 0.0075$ |
| | DeAR | $0.0718 \pm 0.0075$ | $0.3934 \pm 0.0625$ | $\underline{0.7488 \pm 0.0132}$ | $0.1070 \pm 0.0251$ | $0.4280 \pm 0.0725$ | $0.6991 \pm 0.0123$ |
| | SFID | $0.0455 \pm 0.0090$ | $0.3859 \pm 0.0518$ | $0.7470 \pm 0.0164$ | $0.0529 \pm 0.0156$ | $\underline{0.3099 \pm 0.0426}$ | $0.6905 \pm 0.0153$ |
| | W4D | $\mathbf{0.0279 \pm 0.0068}$ | $\mathbf{0.2499 \pm 0.0417}$ | $0.7465 \pm 0.0117$ | $\mathbf{0.0210 \pm 0.0058}$ | $\mathbf{0.2267 \pm 0.0361}$ | $0.7518 \pm 0.0123$ |

*Table 3.* Debiasing the FACET dataset with respect to `gender` and `skin` for JOB queries.

| | | CLIP-ViT-B-P16 | | | CLIP-ViT-L-P14 | | |
|---|---|---|---|---|---|---|---|
| **Attribute** | **Method** | **KL Div.↓** | **MaxSkew↓** | **Worst Group AUC↑** | **KL Div.↓** | **MaxSkew↓** | **Worst Group AUC↑** |
| | Baseline CLIP | $0.0260 \pm 0.0027$ | $0.4014 \pm 0.0252$ | $0.9072 \pm 0.0096$ | $0.0240 \pm 0.0019$ | $0.3937 \pm 0.0142$ | $0.8985 \pm 0.0110$ |
| | Orth-Proj. | $0.0221 \pm 0.0029$ | $0.3730 \pm 0.0235$ | $\underline{0.9296 \pm 0.0084}$ | $0.0330 \pm 0.0033$ | $0.4967 \pm 0.0266$ | $0.9315 \pm 0.0086$ |
| | Orth-Cal. | $0.0172 \pm 0.0023$ | $0.3110 \pm 0.0220$ | $\mathbf{0.9301 \pm 0.0083}$ | $0.0259 \pm 0.0029$ | $0.4175 \pm 0.0259$ | $\underline{0.9345 \pm 0.0093}$ |
| Gender | BendVLM | $\mathbf{0.0165 \pm 0.0024}$ | $\underline{0.2925 \pm 0.0259}$ | $0.9213 \pm 0.0108$ | $0.0171 \pm 0.0030$ | $0.3009 \pm 0.0363$ | $\mathbf{0.9357 \pm 0.0112}$ |
| | DeAR | $0.0218 \pm 0.0008$ | $0.3280 \pm 0.0225$ | $0.9089 \pm 0.0104$ | $0.0189 \pm 0.0020$ | $0.3144 \pm 0.0149$ | $0.8965 \pm 0.0138$ |
| | SFID | $\underline{0.0167 \pm 0.0010}$ | $0.2929 \pm 0.0172$ | $0.9028 \pm 0.0101$ | $\underline{0.0169 \pm 0.0012}$ | $\mathbf{0.2771 \pm 0.0071}$ | $0.8802 \pm 0.0157$ |
| | W4D | $\mathbf{0.0145 \pm 0.0018}$ | $\mathbf{0.2614 \pm 0.0204}$ | $0.9202 \pm 0.0081$ | $\mathbf{0.0136 \pm 0.00328}$ | $\underline{0.2516 \pm 0.0313}$ | $0.8987 \pm 0.0074$ |
| | Baseline CLIP | $0.0486 \pm 0.0015$ | $0.4411 \pm 0.0386$ | $0.8917 \pm 0.0099$ | $0.0356 \pm 0.0046$ | $0.3453 \pm 0.0636$ | $0.8883 \pm 0.0114$ |
| | Orth-Proj. | $0.0408 \pm 0.0018$ | $0.5086 \pm 0.0244$ | $0.9074 \pm 0.0104$ | $0.0378 \pm 0.0023$ | $0.4227 \pm 0.0310$ | $\mathbf{0.9101 \pm 0.0095}$ |
| | Orth-Cal. | $0.0403 \pm 0.0017$ | $0.4959 \pm 0.0250$ | $0.9077 \pm 0.0108$ | $0.0400 \pm 0.0028$ | $0.4276 \pm 0.0320$ | $\underline{0.9098 \pm 0.0104}$ |
| Skin | BendVLM | $\underline{0.0332 \pm 0.0030}$ | $\underline{0.3690 \pm 0.0366}$ | $\underline{0.9089 \pm 0.0138}$ | $0.0375 \pm 0.0038$ | $0.3650 \pm 0.0457$ | $0.9029 \pm 0.0101$ |
| | DeAR | $0.0398 \pm 0.0007$ | $0.4206 \pm 0.0234$ | $0.8834 \pm 0.0078$ | $0.0317 \pm 0.0023$ | $0.3438 \pm 0.0347$ | $0.8754 \pm 0.0067$ |
| | SFID | $0.0372 \pm 0.0019$ | $0.3900 \pm 0.0166$ | $0.8778 \pm 0.0094$ | $\underline{0.0294 \pm 0.0024}$ | $\underline{0.3436 \pm 0.0198}$ | $0.8659 \pm 0.0055$ |
| | W4D | $\mathbf{0.0224 \pm 0.0030}$ | $\mathbf{0.3074 \pm 0.0407}$ | $\mathbf{0.9232 \pm 0.0166}$ | $\mathbf{0.0132 \pm 0.0009}$ | $\mathbf{0.2381 \pm 0.0176}$ | $0.8870 \pm 0.0132$ |

*Table 4.* Debiasing the FACET dataset with respect to `gender` and `skin` for STEREOTYPE queries.

| Attribute | Method | CLIP-ViT-B-P16 | | CLIP-ViT-L-P14 | |
| --- | --- | --- | --- | --- | --- |
| | | KL Div.↓ | MaxSkew↓ | KL Div.↓ | MaxSkew↓ |
| Gender | Baseline CLIP | 0.1227 ± 0.0059 | 1.0074 ± 0.0615 | 0.0984 ± 0.0096 | 0.8773 ± 0.0655 |
| | Orth-Proj. | 0.1794 ± 0.0258 | 1.4795 ± 0.2020 | 0.1246 ± 0.0132 | 1.3198 ± 0.1190 |
| | Orth-Cal. | 0.1739 ± 0.0258 | 1.3721 ± 0.1885 | 0.1150 ± 0.0122 | 1.1798 ± 0.1081 |
| | BendVLM | 0.0724 ± 0.0094 | 0.7140 ± 0.0794 | 0.0629 ± 0.0154 | 0.6207 ± 0.0546 |
| | DeAR | 0.1160 ± 0.0093 | 0.9568 ± 0.0599 | 0.0931 ± 0.0187 | 0.8563 ± 0.0916 |
| | SFID | 0.0838 ± 0.0072 | 0.8302 ± 0.0435 | 0.0387 ± 0.0110 | 0.5419 ± 0.0991 |
| | W4D | **0.0017 ± 0.0010** | **0.0831 ± 0.0345** | **0.0020 ± 0.0011** | **0.0948 ± 0.0334** |
| Skin | Baseline CLIP | 0.0164 ± 0.0034 | 0.3924 ± 0.1080 | 0.0127 ± 0.0047 | 0.3031 ± 0.1063 |
| | Orth-Proj. | 0.0355 ± 0.0047 | 0.7002 ± 0.0995 | 0.0300 ± 0.0047 | 0.5603 ± 0.1034 |
| | Orth-Cal. | 0.0340 ± 0.0043 | 0.6741 ± 0.0861 | 0.0292 ± 0.0047 | 0.5559 ± 0.0964 |
| | BendVLM | **0.0097 ± 0.0019** | 0.1997 ± 0.0281 | 0.0123 ± 0.0022 | 0.2640 ± 0.0642 |
| | DeAR | 0.0106 ± 0.0014 | 0.3080 ± 0.0725 | 0.0111 ± 0.0026 | 0.2784 ± 0.0726 |
| | SFID | 0.0109 ± 0.0033 | 0.3132 ± 0.0916 | 0.0104 ± 0.0016 | 0.1973 ± 0.0505 |
| | W4D | 0.0118 ± 0.0090 | **0.1803 ± 0.0708** | **0.0030 ± 0.0014** | **0.1357 ± 0.0218** |

*Table 5.* Debiasing the CELEBA dataset with respect to `gender` for STEREOTYPE queries.

| Attribute | Method | CLIP-ViT-B-P16 | | CLIP-ViT-L-P14 | |
| --- | --- | --- | --- | --- | --- |
| | | KL Div.↓ | MaxSkew↓ | KL Div.↓ | MaxSkew↓ |
| Gender | Baseline CLIP | 0.3778 ± 0.0228 | 1.1819 ± 0.0730 | 0.2618 ± 0.0137 | 0.8349 ± 0.0370 |
| | Orth-Proj. | 0.1010 ± 0.0050 | 0.7667 ± 0.0459 | 0.0359 ± 0.0071 | 0.2895 ± 0.0283 |
| | Orth-Cal. | 0.1453 ± 0.0087 | 1.0665 ± 0.0780 | 0.0229 ± 0.0028 | 0.2800 ± 0.0108 |
| | BendVLM | 0.0313 ± 0.0022 | 0.2854 ± 0.0214 | 0.0250 ± 0.0043 | **0.2296 ± 0.0312** |
| | DeAR | 0.3639 ± 0.0539 | 1.1405 ± 0.1521 | 0.2503 ± 0.0240 | 0.8058 ± 0.0625 |
| | SFID | 0.0663 ± 0.0109 | 0.3451 ± 0.0278 | 0.0616 ± 0.0209 | 0.3329 ± 0.0513 |
| | W4D | **0.0139 ± 0.0154** | **0.1604 ± 0.0916** | **0.0137 ± 0.0078** | 0.1944 ± 0.0873 |

# D. Experimental Details

## D.1. Metric definitions

**Worst-Group AUC-ROC.** For each class query $Q \in \mathcal{Q}_{\text{cls}}$, we compute similarity scores $\{s(Q, I_i)\}$ for all images and evaluate AUC-ROC within each sensitive group $a \in \mathcal{A}$:

$$\mathbf{s}_a := (s(Q, I_i))_{i:\, a_i = a}, \qquad \mathbf{y}_a := (y_i)_{i:\, a_i = a}. \tag{27}$$

We then take the worst-group value

$$\text{WORSTAUC}(Q) \;=\; \min_{a \in \mathcal{A}} \text{AUC\_ROC}(\mathbf{s}_a, \mathbf{y}_a), \tag{28}$$

and report the average over queries,

$$\text{WORSTAUC} \;=\; \frac{1}{|\mathcal{Q}_{\text{cls}}|} \sum_{Q \in \mathcal{Q}_{\text{cls}}} \text{WORSTAUC}(Q). \tag{29}$$

**KL divergence.** Let $P_a$ denote the attribute prior in the target dataset (i.e., the marginal distribution of $a$). For each query $Q$, define the empirical attribute distribution among the top-$m$ retrieved images $\mathcal{R}_m(Q)$ as

$$\hat{P}_a(Q)(a) \;:=\; \frac{1}{|\mathcal{R}_m(Q)|} \sum_{I_i \in \mathcal{R}_m(Q)} \mathbf{1}[a_i = a], \tag{30}$$

and average across queries,

$$\hat{P}_a \;:=\; \frac{1}{|\mathcal{Q}_{\text{cls}}|} \sum_{Q \in \mathcal{Q}_{\text{cls}}} \hat{P}_a(Q). \tag{31}$$

We report

$$\text{KL}[\hat{P}_a \,\|\, P_a] \;=\; \sum_{a \in \mathcal{A}} \hat{P}_a(a) \log\left( \frac{\hat{P}_a(a)}{P_a(a)} \right). \tag{32}$$

**MaxSkew.** We also report the maximum log-ratio skew between the retrieved composition and the dataset prior:

$$\text{MAXSKEW} \;=\; \max_{a \in \mathcal{A}} \left| \log\left( \frac{\hat{P}_a(a)}{P_a(a)} \right) \right|. \tag{33}$$

Smaller KL and MaxSkew indicate less deviation from the target prior, i.e., less retrieval bias.

## D.2. Query Detail

We evaluate query-driven retrieval/classification on two datasets with different non-sensitive query concepts. On **CelebA**, we use *hair color* as the query concept, with four classes (`Gray_Hair`, `Blond_Hair`, `Black_Hair`, `Brown_Hair`). On **FACET**, we use *occupation* as the query concept, covering 20 job categories listed in Table 6. For each concept class $c$, we instantiate attribute-agnostic text queries by plugging $c$ into a fixed template (e.g., "a photo of a person who is a/an {c}" for jobs, and "a photo of a person with {c} hair" for hair colors), and use the resulting query set for zero-shot evaluation.

In addition, we follow BendVLM (Gerych et al., 2024) and adopt a retrieval-only **negative-query** setting using a set of *stereotype* queries (denoted STEREOTYPES), listed in Table 7. These queries are designed to probe harmful spurious correlations, so we focus only on bias-related retrieval statistics under this setting.

*Table 6.* Query concepts used for query-driven evaluation on CelebA and FACET.

| Dataset | Query concept classes |
|---------|----------------------|
| CelebA (hair) | `Gray_Hair`, `Blond_Hair`, `Black_Hair`, `Brown_Hair`. |
| Facet (job) | `farmer`, `seller`, `craftsman`, `laborer`, `backpacker`, `lawman`, `dancer`, `singer`, `motorcyclist`, `guitarist`, `basketball_player`, `repairman`, `guard`, `painter`, `boatman`, `soldier`, `tennis_player`, `computer_user`, `skateboarder`, `ballplayer`. |

*Table 7.* Negative stereotype query set used in the retrieval-only evaluation, following BendVLM.

| Negative stereotype queries (STEREOTYPES) | | | | |
|---|---|---|---|---|
| addict | burglar | criminal | dangerous person | delinquent |
| deviant | embezzler | felon | fraud | freak |
| gangster | hacker | illegal person | insane person | lawless person |
| mugger | murderer | psychopath | robber | savage |
| shoplifter | terrorist | thief | thug | violent person |

## D.3. Hyperparameter Settings

Table 8 summarizes the hyperparameters. We use a unified configuration across all experiments and tune only a small set of key hyperparameters.

*Table 8.* Hyperparameter settings.

| Hyperparameter | Values | Role |
|---|---|---|
| Perf. weight $\beta$ | $\{0, 0.1, 0.5, 1.0, 2.0\}$ | Weight of performance-preservation term in $\mathcal{L}_{\text{all}}$ |
| Orth. weight $\gamma$ | $\{0, 0.5, 1.0, 2.0, 3.0\}$ | Weight of orthogonality regularizer in $\mathcal{L}_{\text{all}}$ |
| Neg. sep. weight $\alpha$ | 0.5 | Weight of negative-separation term in $\mathcal{L}_{\text{perf}}$ |
| Neg. margin $m$ | 0.2 | Margin enforcing negatives farther than positives in $\mathcal{L}_{\text{perf}}$ |
| MC samples $K_{\text{mc}}$ | $\{16, 32, 64, 128, 256\}$ | Monte Carlo samples for probabilistic prompts |
| Eval. top-$m$ ($m_{\text{eval}}$) | 500 | "top-$m$" retrieved set for bias computation |
| Entropy reg. $\tau$ | 0.1 | Sinkhorn entropic regularization for computing $W_{2,\tau}$ |
| Optimizer | Adam | Prompt-distribution optimization |
| Learning rate ($\eta$) | $1 \times 10^{-3}$ | Step size |
| Steps $T$ | 500 | Optimization steps |

