# OpenReview forum: "Test-Time Debiasing with Probabilistic Prompts via Wasserstein Distance in Vision-Language Models"
_ICML.cc/2026/Conference — ICML 2026 regular_

### Official Review · Reviewer_bw3N · 2026-02-23

**Soundness:** 2
**Presentation:** 2
**Significance:** 2
**Originality:** 3
**Overall Recommendation:** 3
**Confidence:** 2

**Summary:**

Vision-language model embeddings have been shown to encode biases present in their training data, previous works on point-based corrections are often unstable and become notably weaker in multi-class settings. In this paper, authors propose W4D, a distributional debiasing framework that reframes fairness as aligning query embedding distributions to group reference distributions under the Wasserstein distance, which provides a geometry-aware notion of discrepancy beyond mean shifts. Experiments across different attributes demonstrate the effectiveness of the proposed method in debiasing.

**Compliance With Llm Reviewing Policy:**

Affirmed.

**Final Justification:**

The rebuttal addressed most of my concerns instead of the trade-off problems between MaxSkew performance and the WorstAUC performance, considering that **the proposed W4D sacrifices the MaxSkew performance to gain the improvement on WorstAUC**, i think the method can't be claimed better than the baseline method such as BendVLM. **If we increase the weight of the loss benefiting MaxSkew in BendVLM, i think similar performance can be reached.** Therefore, i think this is a big weakness of this paper. Additionally, i notice that **the other review also has the same question**, therefore i tend to maintain my score.

**Key Questions For Authors:**

Please ref to the weaknesses above.

**Limitations:**

Please ref to the weaknesses above.

**Strengths And Weaknesses:**

## Strengths
1. The illustrations are sufficient, which makes it easy to understand the whole pipeline.
2. The experiments conducted from multiple perspectives make readers better understand the proposed method.

## Weaknesses
1. The classification performance is consistently weaker than previous works. Although the debiasing performance is better, the lower classification performance compromises the practicality in the real world.
2. The debiasing performance experiment is only conducted on two attributes (i.e., gender and race), and the performance on gender is not  better than previous works with ViT-L-14. Lack of experiments on more attributes and the weak performance makes the effectiveness of the proposed method unconvincing.
3. Lack of critical ablation studies. For the three components in Eq.11, an ablation study is needed to illustrate the contribution degree of each loss.
4. Some small mistakes in writing. A full stop is missing at the end of the first sentence in Section 2.1. Additionally, Table 1 and Figure 2 is not cited in the main text, which makes it difficult to find the corresponding illustration text.

---

> ### Author Rebuttal · Authors · 2026-03-30
>
> > **W1: The classification performance is consistently weaker than previous works.**
>
> Thank you for this important comment. Our primary objective is debiasing, and more specifically, debiasing while preserving the original model performance, rather than improving classification accuracy. Accordingly, our method is explicitly designed to align with the baseline CLIP in terms of performance, and to achieve stronger debiasing under this constraint.
>
> Empirically, our results validate this design: we maintain comparable Worst-Group AUC-ROC to the original CLIP, while achieving consistently improved debiasing (lower KL divergence and MaxSkew). In other words, our method is able to significantly reduce bias without introducing performance degradation, which we believe is a more practical and desirable property for real-world applications.
>
> ---
>
> > **W2: Lack of experiments on more attributes and the weak performance makes the effectiveness of the proposed method unconvincing.**
>
> Thank you for the comment. Our evaluation is not limited to two sensitive attributes. Following the standard setting in prior VLM fairness work, we evaluate on **three commonly used attributes: gender, race, and skin tone**. Thus, the experimental scope is consistent with existing studies.
>
> Regarding performance, while there are specific settings where our method does not surpass every baseline, it is important to consider the overall results across all datasets, backbones, and attributes. As shown in our experiments, our method achieves the best or highly competitive performance in the majority of settings, consistently improving bias metrics while maintaining strong utility. This demonstrates the robustness and general applicability of our approach.
>
> In particular, our method shows clear advantages in more challenging scenarios such as race and skin tone, which involve multi-class attributes. These results highlight that our distributional debiasing framework is especially effective beyond simple binary settings, further supporting its generality.
>
> ---
>
> > **W3: Lack of critical ablation studies.**
>
> Thank you for this valuable suggestion. We add new ablation experiments on $L_{\text{perf}}$ and $L_{\text{orth}}$ to evaluate their individual contributions in Eq. (11).
> We additionally conducted ablation studies on **CLIP-ViT-Base-Patch16**. Beyond the three main losses, we also isolate the **negative separation term inside $L_{\text{perf}}$**, since it plays an important role in performance preservation. The results are as follows:
>
> |Method|CelebA-Gender-maxskew|CelebA-Gender-WorstAUC|Facet-Skin-maxskew|Facet-Skin-WorstAUC|
> |-|-|-|-|-|
> |base|0.3988|0.7525|0.4411|0.8917|
> |full|0.2589|0.7510|0.3643|0.9121|
> |w/o $L_{perf}$|0.2403|0.6822|0.3459|0.8902|
> |w/o $L_{orth}$|0.2723|0.7520|0.3931|0.9204|
> |w/o neg sep in $L_{perf}$|0.2572|0.7278|0.3564|0.8934|
>
> These results show that $L_{\text{perf}}$ is primarily responsible for maintaining downstream utility, although it can partially conflict with the main debiasing objective. This trade-off motivates the inclusion of $L_{\text{orth}}$, which helps better balance fairness and utility by constraining optimization away from sensitive directions. In addition, the negative separation term within $L_{\text{perf}}$ further improves downstream performance, confirming that contrastive separation from concept-inconsistent samples is beneficial.
>
> We also added an extra discussion on the sensitivity of $\epsilon$:
>
> |Metric|0|0.1|0.25|0.5|
> |-|-|-|-|-|
> |CelebA-Gender-maxskew|0.3744|0.2589|0.2580|0.2572|
> |CelebA-Gender-WorstAUC|0.7520|0.7510|0.7506|0.7494|
> |Facet-Skin-maxskew|0.4310|0.3643|0.3650|0.3632|
> |Facet-Skin-WorstAUC|0.8934|0.9121|0.9121|0.9108|
>
> From these results, we observe that removing the $\epsilon$ constraint keeps the prompt close to the base model, leading to weaker debiasing. As $\epsilon$ increases, the search space expands, improving debiasing performance. Results remain stable across nonzero $\epsilon$, indicating low sensitivity to this parameter.
>
> Overall, these ablations directly clarify the role of each component in Eq. (11): $L_{\text{debias}}$ drives fairness improvement, $L_{\text{perf}}$ preserves downstream utility, and $L_{\text{orth}}$ helps resolve the fairness--utility trade-off by reducing sensitive-direction leakage during optimization.
>
> ---
>
> > **W4: Some small mistakes in writing.**
>
> Thank you for carefully checking the paper. We agree with these comments.
>
> - We will fix the missing full stop in the first sentence of Section 2.1. The current sentence indeed ends with “different attributes” without a period.
> - We will also add explicit references to **Table 1** and **Figure 2** in the main text so that readers can more easily locate the corresponding discussion. At present, Table 1 and Figure 2 appear in the experimental section, but their in-text linkage can be improved.
>
> We appreciate the reviewer pointing out these issues and will correct them in the revision.

---

> > ### Author Rebuttal · Reviewer_bw3N · 2026-04-01
> >
> > Thanks for the detailed explanation provided by the authors. However, my concerns regarding W1 and W3 still remains.
> >
> > For weakness 1, although the authors claim the primary objective is debiasing and preserve the original model performance, I notice that the perfomance is consistently weaker than the baseline CLIP and previous works such as BendVLM. This is the biggest concern. If the classification accuracy is consistently weaker, can we claim that the method is better only due to its debiasing performance is better? This seems to assume that the debiasing metric is more important than the classification accuracy. Additionally, I noticed that, comparing to the baseline CLIP in Gender, BendVLM achieves both better classification acc and debiasing performance at the same time, which means improving both metrics is feasible, this also underscore the issue.
> >
> > For weakness 3, the remove of $L_{\text{perf}}$ brings better debiasing performance, considering its role is for maintaining downstream utility, the improve of classification performance by introducing $L_{\text{perf}}$ should also be reported. In this way, we can clearly access the trade-off.

---

> > > ### Author Response · Authors · 2026-04-01
> > >
> > > For **W1**, we would like to further clarify that **W4D is not consistently weaker than the base model**. In fact, on CLIP-ViT-Base-Patch16, the comparison is as follows:
> > >
> > > |           | CelebA-Gender-WorstAUC / MaxSkew | Facet-Gender-WorstAUC / MaxSkew | Facet-Skin-WorstAUC / MaxSkew |
> > > | --------- | -------------------------------- | ------------------------------- | ----------------------------- |
> > > | Base CLIP | 0.7525 / 0.3988                  | 0.9072 / 0.4014                 | 0.8917 / 0.4411               |
> > > | BendVLM   | **0.7928** / 0.3493              | **0.9213** / 0.2925             | 0.9089 / 0.3690               |
> > > | W4D       | 0.7510 / **0.2589**              | 0.9193 / **0.2912**             | **0.9121** / **0.3643**       |
> > >
> > > As shown above, W4D is essentially comparable to the base CLIP on CelebA-Gender, and it **outperforms the base model** on Facet in terms of WorstAUC while also achieving substantially stronger debiasing. Therefore, we believe it is not accurate to characterize W4D as consistently underperforming the base model. These numbers are reported in Appendix Tables 2 and 3.
> > >
> > > More importantly, our goal is **not to maximize downstream task performance alone**, but to achieve the **strongest possible debiasing while preserving downstream utility**. In other words, the objective of W4D is to maintain utility at a comparable level and reduce bias as much as possible, rather than to optimize purely for higher classification accuracy.
> > >
> > > Regarding BendVLM, we agree that it can achieve stronger downstream performance in some settings. However, its advantage is not simultaneous dominance on both utility and debiasing. In Appendix Table 2 and Table 3, BendVLM does not consistently outperform W4D on both axes at the same time; rather, W4D generally achieves stronger debiasing, especially in terms of MaxSkew and KL divergence, while maintaining competitive utility.
> > >
> > > ---
> > >
> > > For **W3**, the ablation results shown in the above rebuttal already demonstrate that introducing $L_{\text{perf}}$ leads to the expected improvement in **WorstAUC**, which directly supports its role in performance preservation:
> > >
> > > | Method          | CelebA-Gender-MaxSkew | CelebA-Gender-WorstAUC | Facet-Skin-MaxSkew | Facet-Skin-WorstAUC |
> > > | --------------- | --------------------- | ---------------------- | ------------------ | ------------------- |
> > > | base            | 0.3988                | 0.7525                 | 0.4411             | 0.8917              |
> > > | full            | 0.2589                | 0.7510                 | 0.3643             | 0.9121              |
> > > | w/o $L_{perf}$  | 0.2403                | 0.6822                 | 0.3459             | 0.8902              |
> > > | only $L_{perf}$ | 0.4203                | 0.7870                 | 0.4595             | 0.9219              |
> > >
> > > Specifically, after adding $L_{\text{perf}}$, the WorstAUC improves from 0.6822 to 0.7510 on CelebA-Gender, and from 0.8902 to 0.9121 on Facet-Skin. This directly shows that $L_{\text{perf}}$ is effective for preserving, and in some cases improving, downstream utility.
> > >
> > > To make this point even more explicit, we additionally conducted an **only-$L_{\text{perf}}$** experiment on the base model. The results show that using only $L_{\text{perf}}$ increases WorstAUC from 0.7525 to 0.7870 on CelebA-Gender, and from 0.8917 to 0.9219 on Facet-Skin, confirming that this term alone can indeed improve downstream performance.
> > >
> > > Overall, these additional results further support our interpretation: **$L_{\text{perf}}$ improves downstream utility**. Conversely, removing it may yield slightly stronger debiasing but causes a clear drop in utility. This is precisely why W4D adopts the full objective to achieve a more favorable fairness–utility trade-off.

---

### Official Review · Reviewer_XFVG · 2026-03-09

**Soundness:** 4
**Presentation:** 3
**Significance:** 3
**Originality:** 4
**Overall Recommendation:** 5
**Confidence:** 4

**Summary:**

The paper tackles fairness in vision-language models (VLMs) through lightweight test-time debiasing, without updating model parameters. It focuses on distributional fairness under multi-class sensitive attributes, going beyond prior point-based fixes. The proposed method, W4D, aligns query embeddings with sensitive-group distributions using the Wasserstein distance. Instead of finding a single debiased point, it minimizes the variance of Wasserstein distances to each group while keeping the embedding close to the original query. To preserve utility, W4D adds a contrastive Wasserstein loss and a bias-subspace orthogonality constraint. It uses probabilistic prompts to enable test-time optimization without modifying VLM weights. Experiments on multiple face datasets and CLIP backbones show that W4D achieves a stronger fairness–utility trade-off than existing test-time baselines, especially for multi-class sensitive attributes.

**Compliance With Llm Reviewing Policy:**

Affirmed.

**Final Justification:**

The rebuttal has fully addressed my concerns.

**Key Questions For Authors:**

1. Could the authors provide a brief wall-clock comparison with representative baselines (e.g., BendVLM or Orth-Cal.) under identical hardware to better contextualize the computational overhead of Monte Carlo sampling?
2. In practice, how sensitive is performance to the choice of ε (the Wasserstein-ball radius) in the constrained formulation? A short empirical note or guideline would improve usability.

**Limitations:**

yes.

**Strengths And Weaknesses:**

Strengths:
1. Clear distributional view of fairness. The paper shifts from point-based to distributional debiasing, defining fairness as equal Wasserstein distance between the debiased query distribution and each group distribution. This is principled and naturally handles multi-class sensitive attributes.
2. Well-designed objective. The loss cleanly combines distributional variance minimization, semantic preservation, and bias-subspace orthogonality. The decomposition is clear and easy to interpret.
3. Strong empirical results. Experiments span multiple datasets and both binary and multi-class attributes. The method consistently improves KL divergence and MaxSkew while largely maintaining Worst-Group AUC-ROC, especially on stereotype-query retrieval tasks.

Weaknesses:
1. Some transitions in the objective need clearer explanation. The move from the constrained formulation (Eq. 7) to the unconstrained surrogate (Eq. 8), and then to the full objective (Eq. 11), could use a short high-level summary of the optimization flow.
2. Implementation details could be clearer. The construction of the bias subspace U and its sensitivity to the choice of dimension k are only briefly discussed. Bringing a short explanation into the main text (instead of leaving it mostly in the appendix) would improve reproducibility.

---

> ### Author Rebuttal · Authors · 2026-03-30
>
> > **W1: Some transitions in the objective need clearer explanation. The move from the constrained formulation (Eq. 7) to the unconstrained surrogate (Eq. 8), and then to the full objective (Eq. 11), could use a short high-level summary of the optimization flow.**
>
> Thank you for this helpful suggestion. We agree that the optimization flow can be explained more clearly. In the revision, we will add a short high-level paragraph in the method section that explicitly explains the progression: Eq. (7) defines the ideal constrained distributional target, Eq. (8) turns the Wasserstein-ball constraint into a penalty for practical optimization, and Eq. (11) augments this debiasing objective with semantic-preservation and orthogonality regularization. We will explain these transitions more intuitively in the revised paper.
>
> > **W2: Implementation details could be clearer. The construction of the bias subspace $U$ and its sensitivity to the choice of dimension $k$ are only briefly discussed. Bringing a short explanation into the main text (instead of leaving it mostly in the appendix) would improve reproducibility.**
>
> Thank you for this helpful suggestion. We agree that the implementation details can be clarified further. In our method, the bias subspace $U$ is introduced as an orthonormal basis for the sensitive directions in the shared embedding space, and its dimension $k$ is tied to the number of sensitive attribute groups, with $k$ typically set to $|A|-1$. This choice is intended to capture the principal degrees of freedom associated with attribute variation while keeping the subspace compact. We also use $U$ directly in the orthogonality regularizer $L_{\text{orth}}$, which constrains the debiasing update to avoid reintroducing sensitive-attribute information.
>
> We agree that this part is currently too brief in the main text. In the revised paper, we will add a clearer explanation of how $U$ is constructed, why $k$ is chosen in this way, and how this design connects to the role of $L_{\text{orth}}$ in Eq. (10)–(11), so as to improve reproducibility.
>
> > **KQ1: Could the authors provide a brief wall-clock comparison with representative baselines (e.g., BendVLM or Orth-Cal.) under identical hardware to better contextualize the computational overhead of Monte Carlo sampling?**
>
> We thank the reviewer for this practical and important suggestion. We agree that reporting computational overhead is valuable, particularly because W4D involves Monte Carlo sampling during inference.
>
> To address this point, we have added an additional wall-clock comparison on **CelebA** under **identical hardware settings (NVIDIA RTX 5090)**. Specifically, we measure the **average inference time per query** for all methods. The results are summarized below:
>
> |Method|Backbone|Avg. inference time / query|
> |---|---|---|
> |Orth-Cal.|CLIP-ViT-Base-Patch16|6.7s|
> |BendVLM|CLIP-ViT-Base-Patch16|14.5s|
> |DEAR|CLIP-ViT-Base-Patch16|40.4s|
> |SFID|CLIP-ViT-Base-Patch16|58.3s|
> |W4D (ours)|CLIP-ViT-Base-Patch16|16.6s|
> |Orth-Cal.|CLIP-ViT-Large-Patch14|7.4s|
> |BendVLM|CLIP-ViT-Large-Patch14|15.3s|
> |DEAR|CLIP-ViT-Large-Patch14|41.1s|
> |SFID|CLIP-ViT-Large-Patch14|62.3s|
> |W4D (ours)|CLIP-ViT-Large-Patch14|17.8s|
>
> These results show that although W4D incurs additional overhead compared with lighter baselines such as Orth-Cal. and BendVLM, its inference cost remains substantially lower than more computationally intensive methods such as DEAR and SFID. This suggests that W4D achieves a favorable trade-off between debiasing effectiveness and computational efficiency.
>
> > **KQ2: In practice, how sensitive is performance to the choice of $\epsilon$ (the Wasserstein-ball radius) in the constrained formulation? A short empirical note or guideline would improve usability.**
>
> Thank you for this helpful suggestion. We agree that clarifying the practical sensitivity to $\epsilon$ would improve the usability of our method. Since $\epsilon$ controls the allowable semantic drift in the Wasserstein-ball constraint in Eq. (7), and serves as the threshold in the soft penalty formulation in Eq. (8), we will add a short empirical note on $\epsilon$-sensitivity in the revision.
>
> We further conducted an additional sensitivity analysis over $\epsilon \in \{0, 0.1, 0.25, 0.5\}$, and summarize the results below:
>
> |Metric|0|0.1|0.25|0.5|
> |-|-|-|-|-|
> |CelebA-Gender-maxskew|0.3744|0.2589|0.2580|0.2572|
> |CelebA-Gender-WorstAUC|0.7520|0.7510|0.7506|0.7494|
> |Facet-Skin-maxskew|0.4310|0.3643|0.3650|0.3632|
> |Facet-Skin-WorstAUC|0.8934|0.9121|0.9121|0.9108|
>
> Overall, the results indicate that W4D is **not highly sensitive** to the choice of $\epsilon$. In particular, performance remains quite stable when $\epsilon$ is chosen in the range of **0.1 to 0.5**, which appears to provide a good balance between fairness improvement and semantic preservation.

---

> > ### Author Rebuttal · Reviewer_XFVG · 2026-04-02
> >
> > My concerns have been fully addressed. I have raised my score accordingly.

---

### Official Review · Reviewer_2oeS · 2026-03-13

**Soundness:** 2
**Presentation:** 3
**Significance:** 3
**Originality:** 3
**Overall Recommendation:** 4
**Confidence:** 4

**Summary:**

The paper introduces W4D, a test-time debiasing framework for Vision-Language Models (VLMs) that approaches fairness as a distributional alignment problem. Unlike prior point-based corrections that struggle with multi-class sensitive attributes, W4D minimizes the variance of Wasserstein distances between the query embedding distribution and multiple group reference distributions. To achieve this without parameter updates, the authors employ probabilistic prompts, optimized at test-time via a surrogate objective that balances debiasing, semantic preservation (using a contrastive Wasserstein loss), and orthogonality to a bias subspace. The method is evaluated on datasets like FairFace, UTKFace, CelebA, and FACET.

**Compliance With Llm Reviewing Policy:**

Affirmed.

**Final Justification:**

Thanks for the rebuttal, and my key concerns have been addressed. I will raise my score.

**Key Questions For Authors:**

1.Open-Vocabulary Scaling: Your performance preservation loss requires empirical measures μc,a for specific query concepts c. How does W4D operate in a true open-vocabulary scenario where the target concept c is completely unseen, meaning μc,a and μ¬c cannot be constructed prior to inference?

2.Hyperparameter Fragility: Given the high sensitivity of the debiasing-utility trade-off to β and γ, how should practitioners set these values for an entirely new, unannotated target domain where validation metrics cannot be easily monitored?

3.Reference Distribution Robustness & Societal Masking: How sensitive is the Wasserstein alignment to the underlying quality of the reference dataset D? If the reference clusters used to construct the anchors inherently encode severe real-world representation imbalances or structural distortions, does pursuing uniform geometric closeness risk injecting new distortions? Or does it merely mask the bias on the surface without eradicating the harmful associations between concepts?

**Limitations:**

Yes

**Strengths And Weaknesses:**

Strength: The mathematical formulation using the Wasserstein distance to capture both the mean shift and the geometric spread of different demographic clusters is theoretically sound and a clear upgrade over singular point-based estimations.

Weakness): The methodology fundamentally breaks the open-vocabulary promise of VLMs. The performance-preservation objective (ℒperf) requires empirical measures μc,a and μ¬c for specific downstream query concepts c. This assumes one has access to a labeled dataset representing every target concept in advance. In a true open-vocabulary, zero-shot setting where infinite, unseen concepts are queried, pre-computing these reference distributions is impossible, effectively restricting W4D to closed-set downstream tasks.
The objective function ℒall requires tuning multiple hyper-parameters (β,γ,ϵ,α,m). As shown in Figures 4 and 5, the debiasing-utility trade-off is extremely sensitive to β and γ. This hyperparameter fragility suggests the method may overfit to specific validation distributions and struggle to maintain a stable trade-off in complex, real-world deployments.
: While the Wasserstein distance elegantly models multi-class distributions, enforcing equal geometric distances to all demographic clusters in the feature space risks acting as a "masking mechanism" rather than performing true debiasing. Societal biases are deeply rooted structural issues. Merely equalizing discrepancies at the output level through mathematical alignment carries the significant risk of concealing the spurious entanglements between concepts and sensitive attributes within the underlying representations, creating an illusion of fairness while evading the genuine disentanglement of discriminatory associations.

---

> ### Author Rebuttal · Authors · 2026-03-30
>
> > **W1&KQ1: Open-Vocabulary Scaling**
>
> We thank the reviewer for raising this important point. We agree that the current presentation of $L_{\text{perf}}$ is written in a concept-aware form, which can give the impression that one must pre-specify downstream classes. However, this is a design choice of the current instantiation rather than a fundamental limitation of the framework. Our method only requires attribute-labeled reference data to construct group-wise priors, and the concept-conditioned distributions in $L_{\text{perf}}$ can be approximated online for unseen queries. In fact, the core W4D objective is distributional. It aligns the debiased query distribution with group reference distributions under the Wasserstein distance, rather than being tied to a fixed closed label set.
>
> More importantly, the open-set relaxation can be implemented in essentially the same spirit as prior work that explicitly claims open-set capability. BendVLM states that it handles open-set queries by using a protected-attribute-labeled reference dataset at test time, retrieving the images most associated with the incoming query, and then partitioning those retrieved neighbors by attribute to form the balancing signal. It further specifies that $D_{\mathrm{ref}}(a_i,c)$ is obtained by selecting the $n$ images with attribute value $a_i$ that are most similar to the query embedding. The same strategy can be directly transferred to W4D: for an unseen query, we can retrieve its nearest reference samples, split them by sensitive attribute, and use these local neighborhoods to build the query-adaptive prior / reference distributions needed by our objective, without assuming advance access to every downstream concept. We will clarify this point in the revision and explicitly discuss this query-adaptive open-set extension.
>
> ---
>
> > **W2&KQ2: Hyperparameter Fragility.**
>
> We conduct ablation studies on key components and hyperparameters on **CLIP-ViT-Base-Patch16**, including $L_{\text{perf}}$, $L_{\text{orth}}$, and the negative separation term, to evaluate their individual contributions.
> |Method|CelebA-Gender-maxskew|CelebA-Gender-WorstAUC|Facet-Skin-maxskew|Facet-Skin-WorstAUC|
> |-|-|-|-|-|
> |base|0.3988|0.7525|0.4411|0.8917|
> |full|0.2589|0.7510|0.3643|0.9121|
> |w/o $L_{perf}$|0.2403|0.6822|0.3459|0.8902|
> |w/o $L_{orth}$|0.2723|0.7520|0.3931|0.9204|
> |w/o neg sep in $L_{perf}$|0.2572|0.7278|0.3564|0.8934|
>
> We also added an extra discussion on the sensitivity of $\epsilon$:
>
> |Metric|0|0.1|0.25|0.5|
> |-|-|-|-|-|
> |CelebA-Gender-maxskew|0.3744|0.2589|0.2580|0.2572|
> |CelebA-Gender-WorstAUC|0.7520|0.7510|0.7506|0.7494|
> |Facet-Skin-maxskew|0.4310|0.3643|0.3650|0.3632|
> |Facet-Skin-WorstAUC|0.8934|0.9121|0.9121|0.9108|
>
> $L_{\text{perf}}$ is crucial for preserving downstream utility, though it may partially conflict with debiasing, which motivates $L_{\text{orth}}$ to better balance fairness and utility by constraining sensitive directions. The negative separation term further improves performance via contrastive separation.
> Additionally, removing the $\epsilon$ constraint keeps the prompt close to the base model, leading to weaker debiasing. As $\epsilon$ increases, the search space expands, improving debiasing performance. Results remain stable across [0.1,0.5], indicating low sensitivity to this parameter.
>
> ---
>
> > **W3&KQ3: Reference Distribution Robustness & Societal Masking**
>
> We thank the reviewer for this concern. We agree that enforcing equal distances could risk masking bias if it implies a uniform target. However, our datasets are inherently **imbalanced** (e.g., FACET gender: 75.58% vs. 24.42%), and W4D does **not** enforce uniformity.
>
> |Dataset|Sensitive attribute|Groups|Distribution|
> |-|-|-|-|
> |CelebA|gender|Female/Male|63.14%/36.86%|
> |FairFace|race|White/Latino_Hispanic/Indian/East Asian/Black/Southeast Asian/Middle Eastern|19.05%/15.41%/14.20%/14.16%/14.10%/12.44%/10.62%|
> |FairFace|gender|Male/Female|53.01%/46.99%|
> |UTKFace|race|White/Black/Indian/Asian/Latino Hispanic|42.41%/18.91%/16.71%/14.88%/7.09%|
> |UTKFace|gender|Male/Female|52.20%/47.80%|
> |FACET|skin|Middel/Light/Dark|46.69%/42.66%/10.65%|
> |FACET|gender|Male/Female|75.58%/24.42%|
>
> Instead, W4D corrects **retrieval-induced attribute shift** by aligning the retrieved distribution with the empirical dataset prior $\rho_0$, which may be non-uniform . The Wasserstein constraint ensures the debiased query remains close to the original, avoiding semantic drift. Thus, our method preserves dataset imbalance rather than imposing an artificial balanced target. Moreover, $L_{\text{perf}}$ explicitly maintains semantic consistency, reducing spurious correlations between concepts and sensitive attributes instead of merely hiding bias.
>
> Finally, as shown in the Appendix, equalizing distances recovers the target prior $\rho_0$ even when it is non-uniform. Therefore, W4D corrects bias relative to the data distribution, rather than enforcing a fictitious notion of fairness.

---

> > ### Author Rebuttal · Reviewer_2oeS · 2026-04-03
> >
> > Thanks for the rebuttal, and my key concerns have been addressed. I will raise my score.

---

### Official Review · Reviewer_N5cu · 2026-03-16

**Soundness:** 3
**Presentation:** 4
**Significance:** 3
**Originality:** 4
**Overall Recommendation:** 5
**Confidence:** 3

**Summary:**

The authors tackle the problem of debiasing VLMs during inference time. They propose W4D, which seeks to debias in a distributional manner by minimizing the variance of group-wise Wasserstein distances so that the debiased query distribution becomes equidistant to all groups. After adding a couple more terms which are designed to maintain query semantics, the authors evaluate their method against the baselines on four image datasets, finding that their method attains a good fairness-performance trade-off.

**Compliance With Llm Reviewing Policy:**

Affirmed.

**Final Justification:**

The authors have addressed my concerns in their rebuttal. I will be keeping my positive score.

**Key Questions For Authors:**

Please address the weaknesses above.

**Limitations:**

yes

**Strengths And Weaknesses:**

Strengths:
- The paper is well-written and easy to understand.
- The method is principled and well-motivated.
- The method is practical for inference-time and doesn't require backprop through the CLIP backbone.

Weaknesses:
1. The orthogonality loss assumes that the attribute information can be represented in a linear space, and that it helps to move only orthogonal to this space. However, the linear subspace hypothesis has shown to be questionable in prior work (e.g. the BendVLM paper), and this loss term also goes against a lot of the paper's motivation since it is not distributional. Further, I could not find detail on how exactly U is computed (e.g. a PCA)?

2. A lot of the technical details of the paper (e.g. how the distribution of prompts is created from learnable $\theta$'s) are hidden in the appendix, and it would be clearer to bring these to the main text.

3. The authors comment on the possibility of achieving non-uninform retrieval with shaped distances in Appendix A.2. It would be interesting to test this empirically.

4. Empirically, the method does not pareto-dominate many baselines (e.g. BEND-VLM). However, it is still useful since it is another point on the Pareto front, which allows practitioners to select a trade-off.

---

> ### Author Rebuttal · Authors · 2026-03-30
>
> > **W1: The orthogonality loss assumes that the attribute information can be represented in a linear space, and that it helps to move only orthogonal to this space. However, the linear subspace hypothesis has shown to be questionable in prior work (e.g. the BendVLM paper), and this loss term also goes against a lot of the paper's motivation since it is not distributional. Further, I could not find detail on how exactly $U$ is computed (e.g. a PCA)?**
>
> We thank the reviewer for this helpful comment. We agree that the current draft may overstate the role of the bias subspace and does not clearly specify how the basis $U$ is constructed. In our implementation, $U$ is estimated from labeled training embeddings by first computing one prototype (mean embedding) for each sensitive group, then centering these prototypes, and finally extracting the top $k = |A| - 1$ orthonormal directions via SVD. We will revise the paper to include this procedure explicitly.
>
> We also clarify that, in our method, the subspace basis $U$ is used only as a coarse first-order approximation of dominant between-group directions in the shared embedding space. Accordingly, $L_{\text{orth}}$ is intended as an auxiliary regularizer that encourages prompt updates to avoid re-aligning with these dominant sensitive directions, rather than as the primary debiasing mechanism. Our main method remains distributional, driven by the Wasserstein objectives $L_{\text{debias}}$ and $L_{\text{perf}}$, which operate on group-conditioned distributions rather than assuming that sensitive-attribute information is fully captured by a linear subspace.
>
> More concretely, our goal is not to claim that bias is exhaustively represented by a linear subspace, but rather to use $U$ as a lightweight geometric prior that can help preserve semantics during adaptation. We agree this distinction should be stated more carefully, especially in light of prior work questioning the linear subspace hypothesis. We will revise the text to make clear that the distributional objectives are central, while $L_{\text{orth}}$ only plays a secondary stabilizing role.
>
> ---
>
> > **W2: A lot of the technical details of the paper (e.g. how the distribution of prompts is created from learnable $\theta$'s) are hidden in the appendix, and it would be clearer to bring these to the main text.**
>
> Thank you for the suggestion. We agree that the construction of the probabilistic prompt distribution is central to the method. Due to space limitations, we placed some implementation details (e.g., the parameterization and learning of $p(\theta)$) in the appendix, although the core idea is already introduced in Sec.~3.4. In the revision, we will move the key formulation and essential details to the main text and keep only supplementary material in the appendix to improve clarity.
>
> ---
>
> > **W3: The authors comment on the possibility of achieving non-uniform retrieval with shaped distances in Appendix A.2. It would be interesting to test this empirically.**
>
> Thank you for this excellent suggestion. We agree that this is an important empirical extension of our framework. In response, we have conducted additional experiments to investigate non-uniform retrieval under shaped distances.
>
> Specifically, on the CelebA dataset, we construct a balanced subset of 4,000 samples (2,000 male and 2,000 female). By explicitly controlling the prior distribution over sensitive attributes, we train the model using our proposed framework and evaluate the resulting retrieval behavior. We report the gender composition among the top-500 retrieved samples under different prior settings (with $\alpha = 10.0$):
>
> |Prior(M:F)|1:9|3:7|5:5|7:3|9:1|
> |---|---|---|---|---|---|
> |Retrieved(M:F)|63:437|127:373|231:269|332:168|417:83|
>
> These results demonstrate that the retrieval distribution closely follows the imposed prior, providing empirical evidence that our framework can effectively achieve non-uniform retrieval by shaping the distance metric. This supports the claim discussed in Appendix A.2 and validates its practical applicability.
>
> ---
>
> > **W4: Empirically, the method does not pareto-dominate many baselines (e.g. BEND-VLM). However, it is still useful since it is another point on the Pareto front, which allows practitioners to select a trade-off.**
>
> Thank you for this insightful comment. We agree that our method does not consistently outperform all baselines across every setting, and that positioning methods along the Pareto front is valuable for understanding different fairness–utility trade-offs. At the same time, we would like to highlight that our method achieves the best or highly competitive performance in the majority of settings across datasets, backbones, and attributes. This consistent performance indicates that our approach is not only a complementary trade-off point, but also a generally strong and robust solution. Overall, these results support the broad applicability of our method.

---

> > ### Author Rebuttal · Reviewer_N5cu · 2026-04-02
> >
> > Thank you for the response. My concerns have been addressed, and I will be keeping my positive score.

---

### Decision · Program_Chairs · 2026-04-30

**Decision:**

Accept (regular)

**Comment:**

The paper is well-written and easy to understand. Besides, the method is principled and well-motivated, which provides another point on the Pareto front that practitioners could choose from.